Corrected: Author correction

# Competitive endogenous RNA is an intrinsic component of EMT regulatory circuits and modulates EMT

Yuwei Liu[1,2], Mengzhu Xue[1], Shaowei Du[1,3], Wanwan Feng[4], Ke Zhang[4], Liwen Zhang[2,5,6], Haiyue Liu[1,2], Guoyi Jia[1,3], Lingshuang Wu[7], Xin Hu[8,9], Luonan Chen[5,7,10,11] & Peng Wang [1,2,4,8,9]

The competitive endogenous RNA (ceRNA) hypothesis suggests an intrinsic mechanism to regulate biological processes. However, whether the dynamic changes of ceRNAs can modulate miRNA activities remains controversial. Here, we examine the dynamics of ceRNAs during TGF-β-induced epithelial-to-mesenchymal transition (EMT). We observe that TGFBI, a transcript highly induced during EMT in A549 cells, acts as the ceRNA for miR-21 to modulate EMT. We further identify FN1 as the ceRNA for miR-200c in the canonical SNAIL-ZEB-miR200 circuit in MCF10A cells. Experimental assays and computational simulations demonstrate that the dynamically induced ceRNAs are directly coupled with the canonical double negative feedback loops and are critical to the induction of EMT. These results help to establish the relevance of ceRNA in cancer EMT and suggest that ceRNA is an intrinsic component of the EMT regulatory circuit and may represent a potential target to disrupt EMT during tumorigenesis.

[1] Laboratory of Systems Biology, Shanghai Advanced Research Institute, Chinese Academy of Sciences, 200031 Shanghai, China. [2] University of Chinese Academy of Sciences, 200031 Shanghai, China. [3] School of Life Sciences, Shanghai University, 200031 Shanghai, China. [4] Bio-Med Big Data Center, CAS Key Laboratory of Computational Biology, CAS-MPG Partner Institute for Computational Biology, Shanghai Institute of Nutrition and Health, Shanghai Institutes for Biological Sciences, Chinese Academy of Sciences, 200031 Shanghai, China. [5] Key Laboratory of Systems Biology, Center for Excellence in Molecular Cell Science, Institute of Biochemistry and Cell Biology, Shanghai Institutes for Biological Sciences, Chinese Academy of Sciences, 200031 Shanghai, China. [6] Shanghai Institute of Materia Medica, Chinese Academy of Sciences, 200031 Shanghai, China. [7] School of Life Science and Technology, ShanghaiTech University, 201210 Shanghai, China. [8] Department of Breast Surgery, Key Laboratory of Breast Cancer in Shanghai, Fudan University Shanghai Cancer Center, Fudan University, 200032 Shanghai, China. [9] Precision Cancer Medicine Center, Fudan University Shanghai Cancer Center, 200032 Shanghai, China. [10] Center for Excellence in Animal Evolution and Genetics, Chinese Academy of Sciences, 650223 Kunming, China. [11] Research Center for Brain Science and Brain-Inspired Intelligence, 201210 Shanghai, China. These authors contributed equally: Yuwei Liu, Mengzhu Xue, Shaowei Du. Correspondence and requests for materials should be addressed to X.H. (email: xinhu@fudan.edu.cn) or to L.C. (email: lnchen@sibs.ac.cn) or to P.W. (email: wangpeng@picb.ac.cn)

MicroRNAs (miRNAs) are ubiquitous post-transcriptional regulators that impact RNA stability and the rate of translation by pairing to complementary sites (referred to as miRNA response elements [MREs]) within target RNAs[1–3]. The interaction between miRNAs and their RNA targets is characterized by a many-to-many relationship in which a single miRNA can repress multiple-RNA targets and a single RNA can contain MREs of multiple miRNAs. Hence, ceRNA hypothesis has been proposed; this hypothesis suggests that RNAs can regulate each other by competing for a limited pool of miRNAs[4,5]. Studies have suggested that ceRNA crosstalk may regulate essential biological processes such as cancer[6–10]. However, these studies often lack the absolute quantification of miRNAs and the corresponding ceRNAs[11–13]. Consequently, whether the effects of ceRNA exist under physiological conditions has been challenged[11,13,14]. For example, a recent quantitative study demonstrated that the global MRE changes in hepatocytes are not sufficient to modulate the activity of miR-122, and ceRNA may not be a physiological regulator[13].

Though analyses in hepatocytes have clearly demonstrated that ceRNA does not modulate the activity of miR-122, whether the absence of ceRNA activity can be extended to other conditions has not been examined. miRNAs play essential roles in tumorigenesis and both upregulation and downregulation of miRNAs have been reported in various cancers[11,15]. Interestingly, the downregulation of miRNAs in some cancers could potentially establish an environment that renders ceRNA regulations viable. Moreover, dynamic gene expression changes are commonly observed in cancer[16]. Hence, it is critical to extend the analyses of ceRNAs to dynamic biological processes, which would further clarify the role of ceRNA-based regulation and could offer novel insights into the dynamic miRNA activities underlying key biological processes.

Epithelial-to-mesenchymal transition (EMT) is a fundamental developmental program that has been implicated in metastasis, a detrimental process contributing to more than 90% of cancer-related deaths[17–20]. EMT is associated with dramatic changes in the expression of thousands of genes and is characterized by the downregulation of epithelial markers, such as CDH1, and the upregulation of mesenchymal markers, such as CDH2, VIM, and FN1[21,22]. Because of the crucial role of EMT in development and disease, the circuit regulating dynamic gene expression during EMT has been extensively characterized. The canonical EMT-regulatory circuit comprises a double-negative feedback loop between miR-200 and ZEB1[19,23,24], which displays remarkable hypersensitivity and plays an essential role in regulating EMT in cancer[25].

Interestingly, mathematical modeling of the gene expression changes during EMT has revealed a crucial flaw in the current model[26]. Specifically, miRNAs are stable molecules with an average half-life of approximately 120 h[27]. Indeed, the abundance of miR-200c was only modestly downregulated (~20%) at 96 h into TGF-β-induced EMT in MCF10A cells[26]. Importantly, key EMT markers such as CDH1, CDH2, and ZEB1 display much faster dynamics during EMT in MCF10A cells[26,28]. Consequently, to make the simulated dynamics consistent with experimental observations, the simulation must include a half-life of 5 h for miR-200c[26,28]. Because the mathematical models have incorporated all well-established regulators of EMT such as SNAIL1, ZEB1, miR-34, and miR-200, this discrepancy clearly suggests the existence of unknown mechanisms that modulate the activity of miR-200c prior to its transcriptional repression through ZEB1.

An attractive hypothesis is that the miR-200c MREs are sufficiently upregulated to modulate EMT. Here, we provide a systematic analysis establishing ceRNA as an intrinsic component of the EMT-regulatory circuit. We show that a single ceRNA induced during EMT is directly coupled with the canonical double-negative feedback loop and could modulate the activities of EMT-inhibiting miRNAs. These results resolve a key discrepancy between the established EMT-regulatory circuit and experimental observations, facilitating further studies aimed at establishing dynamically induced ceRNAs as key regulators modulating EMT in cancer.

## Results

**FOXP1 is a critical inducer of EMT in A549 cells**. In a previous study, we profiled the transcriptional dynamics of TGF-β-induced EMT in A549 cells[22]. While the canonical EMT regulators, such as SNAI1/2 or ZEB1/2, are not abundantly expressed in A549 cells, differential expression analyses have suggested that FOXP1, a transcription factor (TF) that plays an important role in regulating embryonic stem cell pluripotency[29], is highly induced and could represent a key EMT regulator in A549 cells (Supplementary Fig. 1A–C). To investigate the role of FOXP1 in EMT, we knocked down FOXP1 in A549 cells undergoing TGF-β-induced EMT using short hairpin RNA. Notably, the knockdown of FOXP1 significantly enhanced the expression of CDH1 and repressed the expression of CDH2 (Fig. 1a, b). Moreover, the loss of FOXP1 substantially reduced the migration and invasion of these cells, diminishing the key characteristic of EMT (Fig. 1c, d). To further demonstrate that FOXP1 is a key inducer of EMT, we overexpressed FOXP1 in A549 cells. Consistent with the results of loss-of-function assays, cells overexpressing FOXP1 gained a mesenchymal-like morphology, lost the expression of CDH1, showed upregulated expression of VIM, and acquired an enhanced ability to migrate and invade (Fig. 1e–h, Supplementary Fig. 1D). Taken together, these data confirmed that FOXP1 is a potent transcriptional inducer of EMT in A549 cells.

**FOXP1 and miR-21 forms a double-negative feedback loop**. Interestingly, FOXP1 expression reached a plateau at 48 h after TGF-β treatment (Supplementary Fig. 1A–C). Because the canonical EMT-regulatory network is characterized by double-negative feedback loops between SNAIL-miR-34 and ZEB-miR-200c, we speculated that miRNAs might also regulate FOXP1 activity in A549 cells to establish equilibrium. To identify potential miRNA regulators of FOXP1, we used deep sequencing (miRNA-seq) to profile miRNA expression during TGF-β-induced EMT and identified 19 and 126 differentially expressed miRNAs at 24 and 96 h into EMT, respectively (Fig. 2a, Supplementary Fig. 2A). We focused on miRNAs that were differentially expressed at 96 h into EMT because FOXP1 expression maintained an equilibrium from 48 to 96 h into EMT. To identify candidate regulatory miRNAs for FOXP1, we examined the overlap between miRNAs differentially expressed at 96 h into EMT and miRNAs predicted to regulate FOXP1 by targetScan[30]. While five miRNAs were identified by both targetScan and the differential expression analysis, four of the five miRNAs (miR-122-5p, miR-129-5p, miR-200b-3p, and miR543) were expressed at low levels (counts per million [CPM] < 10) (Supplementary Fig. 2B). Candidate regulatory miRNAs have been typically identified by changes in relative expression, in which larger changes in the relative expression indicate more significant functions. However, increasing evidence has demonstrated that, for miRNAs, a sufficiently high number of miRNA transcripts in cells is essential for the miRNA to be functional, because a low number of miRNA transcripts (<100/cell) cannot effectively repress their targets owing to the dilution effects of large number of MREs[31]. Using published miRNA absolute qPCR and miRNA-seq data, we extrapolated the absolute copy number of the five miRNAs and observed that only miR-21, a well-established oncomiR, was expressed at >100 copies/cell in A549 cells. Thus, we focused our subsequent analyses on miR-21. Interestingly, the

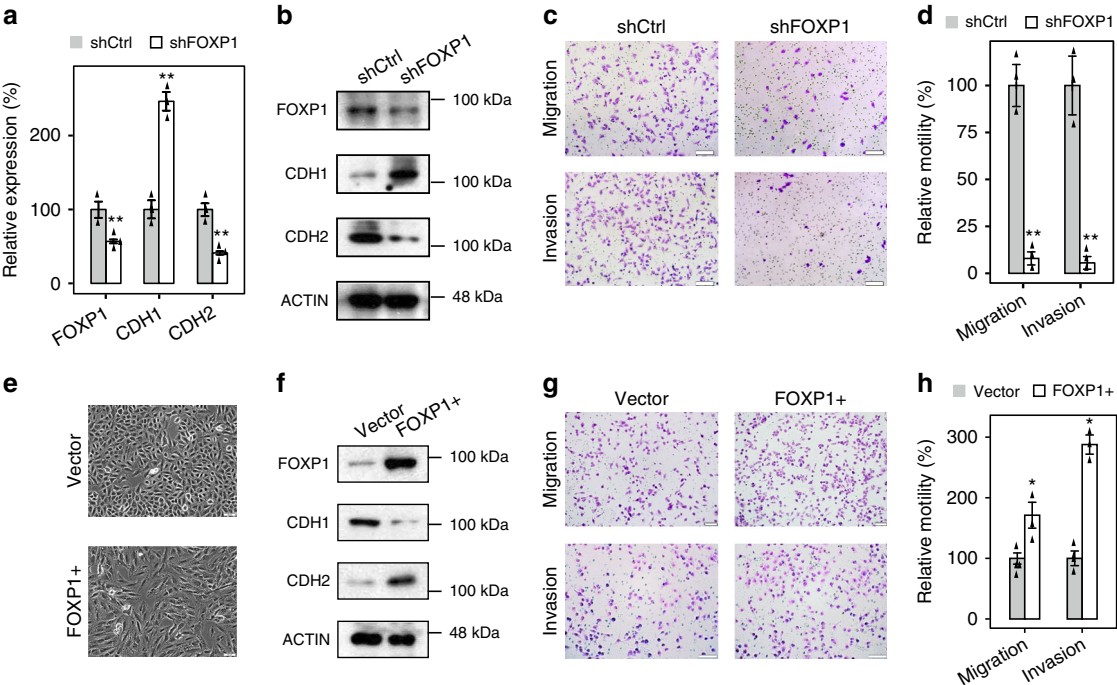

**Fig. 1** FOXP1 is a critical inducer of EMT in A549 cells. **a** Quantitative reverse transcription polymerase chain reaction (qRT-PCR) results showing the levels of gene expression in A549 cells during TGF-β-induced EMT after silencing FOXP1 expression using specific shRNA. $n = 3$. **b** Immunoblotting analysis of the protein abundance of indicated genes in A549 cells undergoing EMT after treatment with an shRNA targeting FOXP1. **c** A549 cells undergoing TGF-β-induced EMT were treated with an shRNA targeting FOXP1 and subjected to a migration assay (upper panel) and an invasion assay (lower panel). Scale bars: 100 μm. **d** The migrated or invaded cells were quantified (bar charts). $n = 6$. **e** Morphology of A549 cells overexpressing FOXP1 or an empty control vector. **f** Immunoblotting analysis of the protein abundance of indicated genes in A549 cells overexpressing FOXP1 or an empty control vector. **g** A549 cells overexpressing FOXP1 or an empty control vector were subjected to a migration assay (upper panel) and an invasion assay (lower panel). Scale bars: 100 μm. **h** The migrated or invaded cells were quantified (bar charts). $n = 6$; error bars indicate the means ± s.d. *$p < 0.01$, determined using the two-tailed Student's $t$ test. Source data are provided as a Source Data file

canonical EMT miRNA, miR-200c, only expressed at very low levels in A549 cells comparing to miR-21 (normalized read counts 43.33 vs. 1,026,301.79). Because the canonical EMT TFs such as SNAIL and ZEB are also expressed at very low levels in A549 cells, we speculated that the canonical SNAIL/ZEB-miR-200c EMT-regulatory circuit is not functional in A549 cells, and FOXP1 and miR-21 are the master molecules to regulate EMT in A549 cells.

Unlike ZEB1, which possesses multiple binding sites for the miR-200 family, FOXP1 only has a single highly conserved binding site for miR-21 (Fig. 2b). To determine whether the miR-21 binding site is functional, we cloned the FOXP1 3′UTR containing the miR-21 binding site into a luciferase reporter and observed that luciferase activity was substantially reduced in the presence of the miR-21 binding site. The inhibitory effect is miR-21-specific because deleting the seed region of the miR-21 binding site or miR-21 knockdown effectively restored luciferase activity (Fig. 2c). We next investigated whether miR-21 is functional during TGF-β-induced EMT. Expectedly, miR-21 knockdown in mesenchymal cells (96 h into TGF-β-induced EMT) significantly enhanced the mRNA and protein abundance of FOXP1, and consequently, further upregulated CDH2 and repressed CDH1 (Fig. 2d, e). The reduction of miR-21 during EMT also significantly enhanced the migration and invasion of these cells, confirming that miR-21 regulates TGF-β-induced EMT in A549 cells (Supplementary Fig. 3). To demonstrate that miR-21 regulates EMT through FOXP1, we mutated the seed region of miR-21 binding site in the 3′UTR of FOXP1 by CRISPR/Cas9 and remeasured the impact of miR-21 knockdown on EMT (Fig. 2f, Supplementary Fig. 3A, B). Critically, the lack of

functional miR-21 binding site completely abolished the impact of miR-21 on the expression of EMT genes (Fig. 2f), and cells' ability to migrate or invade (Supplementary Fig. 3D). Hence, miR-21 regulates EMT strictly through FOXP1 in A549 cells. Moreover, the knockdown of FOXP1 by FOXP1-specific siRNA induced the significant upregulation of miR-21 (Fig. 2g). Taken together, these data confirmed that FOXP1 and miR-21 form the canonical double-negative feedback loop.

**TGFBI regulates EMT as a ceRNA for miR-21**. A canonical theme of EMT-regulatory circuits is that the key miRNAs involved in this process, such as members of the miR-200 family, are downregulated during EMT. Surprisingly, miR-21 demonstrated unorthodox upregulation during TGF-β-induced EMT in A549 cells (Fig. 2a). Although the fold change was modest (1.2 at 96 h into EMT), the increase in the absolute number of miR-21 transcripts was substantial because miR-21 comprised 64.91% of all mappable reads in miRNA-seq at 96 h into EMT. This observation raised the interesting question of how FOXP1 functions in the presence of a large number of miR-21 molecules.

An attractive mechanism to overcome the inhibition of miR-21 is through a ceRNA effect, which requires the presence of sufficient MREs to sequester the miR-21 molecules. To examine this hypothesis, we first analyzed the dynamic change of miR-21 MREs using the time-course RNA-SEQ data during TGF-β-induced EMT. This analysis showed that the miR-21 MREs exhibited a fourfold expansion during TGF-β-induced EMT (Fig. 3a, Supplementary Fig. 4A, Supplementary Tables 1 and 2). Importantly, TGFBI, a gene broadly associated with EMT and with experimentally validated miR-21 binding sites[32], represented

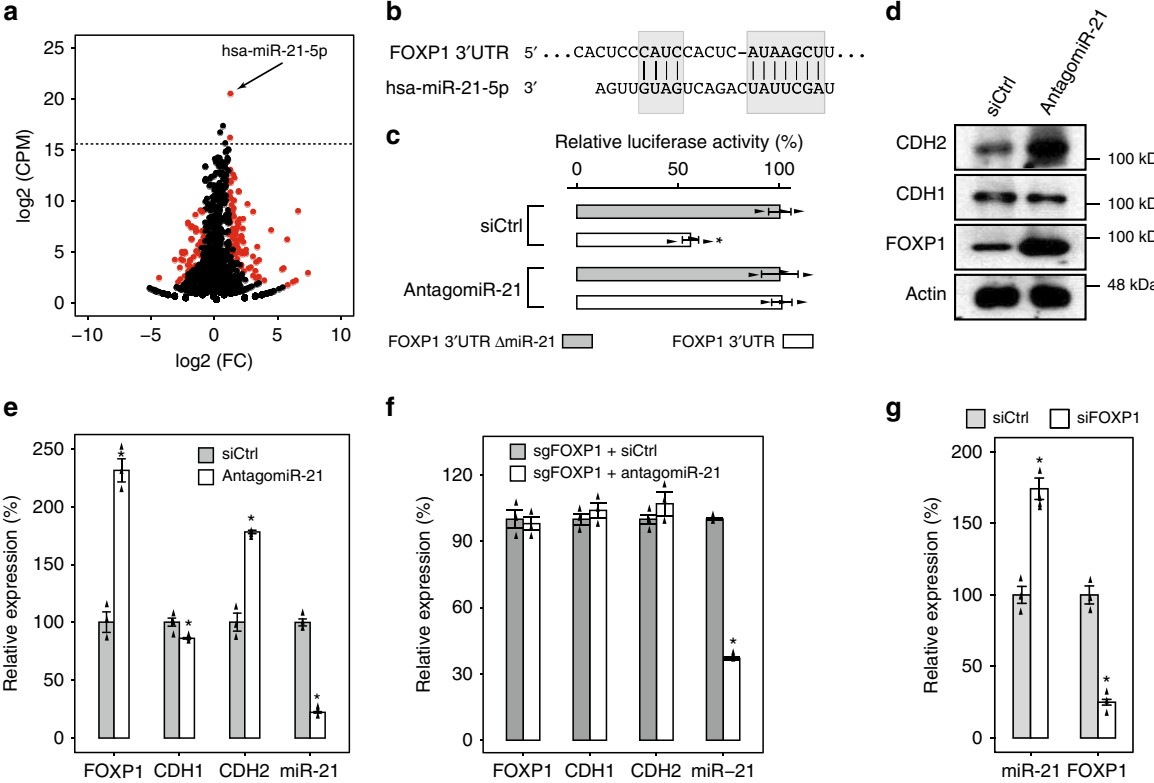

**Fig. 2** FOXP1 and miR-21 form a double-negative feedback loop. **a** Volcano plot showing the differential expression of miRNAs at 96 h into TGF-β-induced EMT in A549 cells. The red dots represent miRNAs with a differential expression FDR < 0.05 and absolute log2-fold change > 1. The horizontal dotted line represents the log2(CPM) corresponding to 100 copies/cell. **b** Graph showing the sequence alignment of FOXP1 3′UTR with miR-21-5p. **c** The results of the luciferase reporter assay were quantified (bar charts). **d** Immunoblotting analysis of the protein abundance of indicated genes in A549 cells during TGF-β-induced EMT after a specific antagomiR was used to silence miR-21 expression. **e** Same as (**d**) for the qRT-PCR assay. **f** qRT-PCR analysis of the indicated genes in A549 cells during TGF-β-induced EMT after a specific antagomiR was used to silence miR-21 expression, using A549 cells whose miR-21 binding site in FOXP1 has been mutated by CRISPR-Cas9. **g** A549 cells undergoing TGF-β-induced EMT were treated with a siRNA targeting FOXP1, and the impact on miR-21 expression was quantified using qRT-PCR. *n* = 3; error bars indicate the means ± s.d. *p < 0.01, determined using a two-tailed Student's *t* test. Source data are provided as a Source Data file

over 90% of all the increased miR-21 MREs in A549 cells during EMT (Supplementary Table 2). Importantly, recent studies have demonstrated that the absolute quantification of miRNAs and their corresponding ceRNAs is critical to establish ceRNA effects[11,13]. Hence, we performed absolute qRT-PCR to quantify the number of miR-21 and TGFBI at several time points during EMT (Supplementary Fig. 4B). As expected, the number of miR-21 was approximately 3.5-fold higher than the number of TGFBI in unstimulated A549 cells, a result that was consistent with reports demonstrating the oncogenic roles of miR-21 in A549 cells[33]. Importantly, TGF-β treatment induced the strong upregulation of TGFBI, and TGFBI outnumbered miR-21 by 1.95-fold (6489 vs. 3323) at 24 h into EMT, suggesting that TGFBI may effectively function as a miR-21 ceRNA to relieve the repression of FOXP1. Moreover, TGFBI demonstrated a modest downregulation late into EMT and outnumbered miR-21 only by 20% at 96 h (5056 vs. 4220). The near-equal amount of TGFBI and miR-21 at 96 h suggested that miR-21 can perform its repressive function during the mesenchymal state, which is consistent with the equilibrium expression of FOXP1 and the results of the miR-21 knockdown experiments shown in Fig. 2.

We next performed computational simulations to calculate the changes of miR-21 binding site occupancy during EMT using estimated number of MREs and miR-21 (Fig. 3b and Supplementary Fig. 4C). Consistent with the dynamic expression patterns of TGFBI and miR-21 during EMT, the 8mer miR-21 binding site occupancy decreased to around 70% at the 12–36 h

period during EMT, and reversed back to 90% at 48 h. The drop was more prominent with MREs estimated by targetScan (average about 20%) than with MREs estimated by pictar (average about 10%), suggesting that different prediction algorithms could have significant impacts on the estimated number of MREs, and consequently, modeled site occupancies. Critically, the miR-21 binding site occupancy was consistently above 90% when the TGFBI MREs were removed from the simulation, suggesting that TGFBI is the key ceRNA molecule during EMT.

We next experimentally examined whether TGFBI indeed exhibited functional ceRNA activities during TGF-β-induced EMT in A549 cells. The results of absolute qPCR revealed that the ceRNA effect of TGFBI is strongest around 24 h into EMT because TGFBI expression peaked around 24 h into EMT. Hence, we knocked down TGFBI using siRNA 24 h prior to TGF-β treatment and analyzed the impact of TGFBI-specific siRNA at 24 h into EMT. Compared with control siRNA, the siRNA specific for TGFBI significantly reduced the expression of FOXP1 and CDH2 and, conversely, enhanced the expression of CDH1 and miR-21, suggesting that TGFBI can directly modulate miR-21 activity through the FOXP1-miR-21 double-negative feedback loop. (Fig. 3c, d). This effect reflected the ceRNA activity of TGFBI because overexpressing the TGFBI 3′UTR containing the miR-21 binding site, but not the control 3′UTR without the miR-21 binding site, could effectively rescue the expression of FOXP1, miR-21, CDH1, and CDH2 (Fig. 3c, d). We further analyzed the ceRNA activity of TGFBI using transwell assays. Expectedly, the

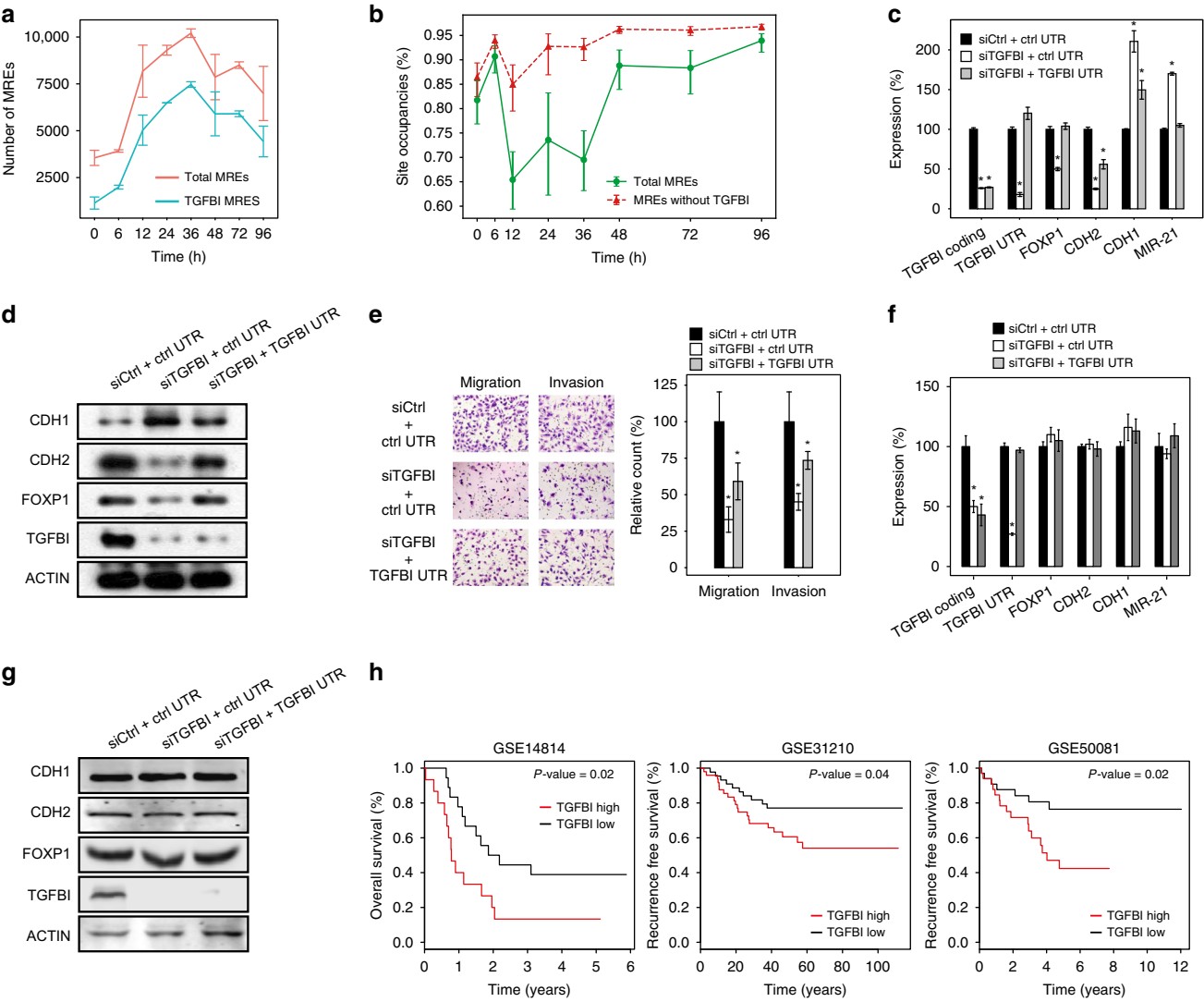

**Fig. 3** TGFBI is a functional ceRNA for miR-21 to regulate EMT. **a** Graph showing the number of miR-21 MREs extrapolated from RNA-seq data during TGF-β-induced EMT in A549 cells using targetScan-based predictions. **b** Graph showing the dynamics of modeled 8-mer miR-21 binding site occupancy during TGF-β-induced EMT in A549 cells using targetScan-based predictions. **c** Graph showing the level of indicated RNAs in A549 cells during TGF-β-induced EMT subjected to the indicated treatments. **d** Same as (**c**) for the immunoblotting analysis. **e** A549 cells undergoing TGF-β-induced EMT were treated as indicated and subjected to a migration assay (left panel) and an invasion assay (right panel). Scale bars: 100 μm. The migrated or invaded cells were quantified (bar charts). n = 6. **f** Graph showing the level of indicated RNAs in A549 cells during TGF-β-induced EMT subjected to the indicated treatments, using cells whose DICER1 has been knocked out with CRISPR-Cas9. **g** Same as (**f**) for the immunoblotting analysis. **h** Kaplan–Meier survival analyses based on the overexpression of TGFBI in three independent lung cancer data sets. n = 3; error bars indicate the means ± s.d. *p < 0.01, determined using a two-tailed Student's t test. Source data are provided as a Source Data file

knockdown of TGFBI significantly reduced the migration and invasion of the cells (Fig. 3e). Consistent with the results of qRT-PCR and immunoblot assays, the reduction of motility could be effectively rescued by overexpressing functional TGFBI 3′UTR containing the miR-21 binding site (Fig. 3e).

A potential pitfall of siRNA-based assays is the saturation of the RISC machinery, which could affect miRNA function indirectly. Thus, we further analyzed the TGFBI ceRNA activity using two additional approaches. We first utilized CRISPR/Cas9 technology to knockout DICER1, which is essential for the biogenesis of miRNAs (Supplementary Fig. 5A, B). As expected, the knockout of DICER1 gene reduced the mature miR-21 abundance by >90% (Supplementary Fig. 5C, D) comparing to wide type controls. Consistent with the hypothesis that the ceRNA effects of TGFBI require functional miRNAs, depleting TGFBI in DICER1-less A549 cells demonstrated no detectable

impacts on the expression of key EMT genes (Fig. 3f, g). Moreover, the introduction of TGFBI 3′UTR with miR-21 binding sites into A549 cells also generated neglectable impacts on key EMT molecules, further confirming that the impact of TGFBI is through the associated ceRNA activities that require mature miRNAs (Fig. 3f, g). Secondly, we mutated the seed region of miR-21 binding site in the 3′UTR of TGFBI with CRISPR/Cas9 technology (Supplementary Fig. 6A–D). We then repeated the ceRNA function assays by depleting TGFBI with specific siRNAs. As expected, no significant impacts on the expression of FOXP1, CDH1, CDH2, and miR-21 were detected in A549 cells lacking functional miR-21 binding sites in the 3′UTR of TGFBI (Supplementary Fig. 6E, F). Interestingly, introducing TGFBI 3′ UTR with functional miR-21 binding sites did generate a small but significant impact on the expression of FOXP1, CDH2, and miR-21 (Supplementary Fig. 6E, F). Taken together, these data

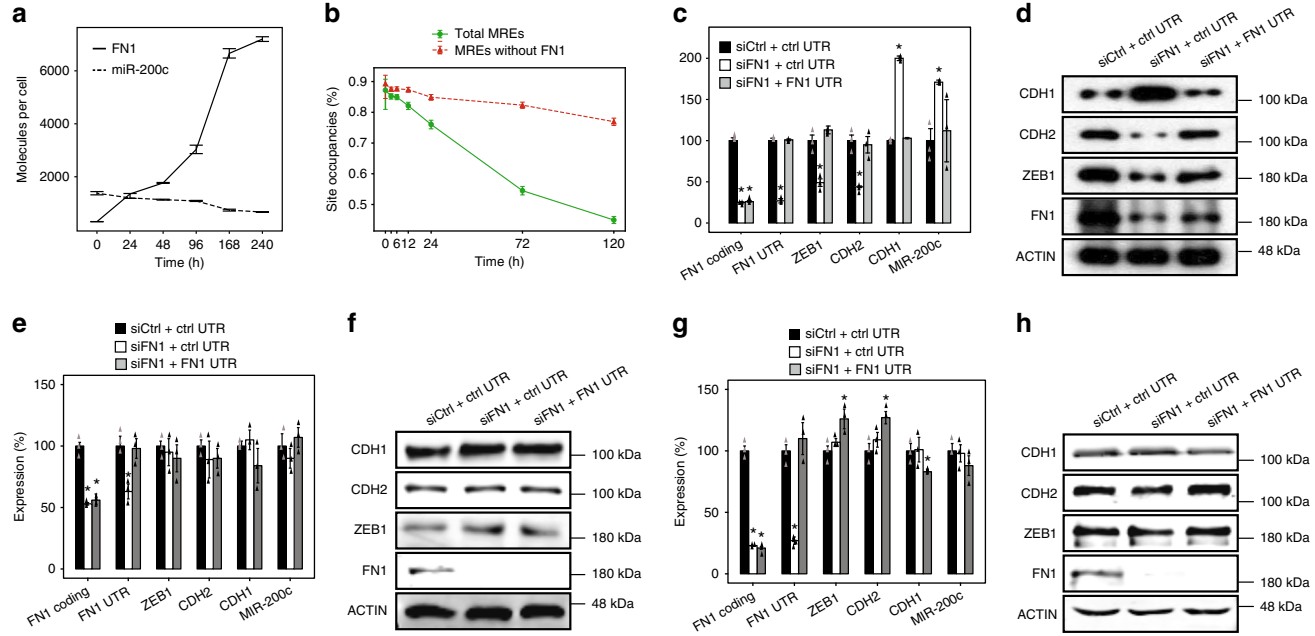

**Fig. 4** FN1 is a functional ceRNA for miR-200c to regulate EMT. **a** Absolute quantification of the numbers of FN1 mRNA and miR-200c transcripts during TGF-β-induced EMT in MCF10A cells. **b** Graph showing the dynamics of modeled 8-mer miR-200c binding site occupancy during TGF-β-induced EMT in MCF10A cells using pictar-based predictions. **c** Graph showing the level of indicated RNAs in MCF10A cells during TGF-β-induced EMT subjected to indicated treatments. **d** Same as (**c**) for immunoblotting analysis. **e** Graph showing the level of indicated RNAs in MCF10A cells during TGF-β-induced EMT subjected to indicated treatments, using cells whose DICER1 has been knocked down. **f** Same as (**e**) for immunoblotting analysis. **g** Graph showing the level of indicated RNAs in MCF10A cells during TGF-β-induced EMT subjected to indicated treatments, using cells whose conserved miR-200c binding site in FN1 3′UTR has been mutated with CRISPR-Cas9. **h** Same as (**g**) for immunoblotting analysis. $n = 3$; error bars indicate the means ± s.d. *$p < 0.01$, determined using a two-tailed Student's $t$ test. Source data are provided as a Source Data file

demonstrated that the observed ceRNA effects of TGFBI operate specifically through the associated miR-21 binding site. Finally, TGFBI knockdown only impacted the expression of genes with experimentally validated miR-21 binding sites, further demonstrating the specificity of the ceRNA effects associated with TGFBI (Supplementary Fig. 6G, H).

We next examined whether the key TFs controlling EMT regulate TGFBI expression. Previously, we established that three master TFs, ETS2, HNF4A, and JUNB, synergistically regulate EMT in A549 cells. As expected, the knockdown of ETS2, HNF4A, or JUNB significantly reduced TGFBI expression, confirming that the master TFs for EMT positively regulate TGFBI expression (supplementary Fig. 6I, J). Moreover, the knockdown of FOXP1 significantly enhanced TGFBI expression, which was consistent with the observed downregulation of TGFBI late into EMT (Supplementary Fig. 6I, J). Taken together, these data suggested that TGFBI, the key ceRNA regulating EMT in A549 cells, is directly coupled with the FOXP1-miR-21 double-negative feedback loop and represents an intrinsic component of the EMT-regulatory circuits.

Because EMT is a crucial step of metastasis, it is of great interest to examine whether TGFBI expression is associated with the clinical outcome of cancer patients. We then performed Kaplan–Meier survival analyses on three independent lung cancer data sets. Reassuringly, the overexpression of TGFBI was consistently associated with a poor clinical outcome in tested sets (Fig. 3h), suggesting that TGFBI may serve as a novel target to intervene metastasis.

**FN1 regulates EMT as a ceRNA for miR-200c.** To demonstrate that the coupling of ceRNA with double-negative feedback loops is a conserved principle underlying EMT-regulatory circuits, we

next examined whether ceRNA also represents a crucial component in the canonical ZEB-miR200 regulatory model of EMT using MCF10A cells. We first used publicly available time-course microarray data to examine whether the MREs for miR-200c, the miRNA that plays a key role in regulating EMT in MCF10A cells, demonstrate significant dynamics during EMT (Supplementary Fig. 7A, B, Supplementary Tables 3 and 4). Strikingly, we observed a pattern analogous to that observed in A549 cells: the MREs for miR-200c displayed significant upregulation, and a single RNA, FN1, represented over 90% of all increased MREs during EMT in MCF10A cells at 72 h into EMT (Supplementary Fig. 7A, B, Supplementary Table 4). Importantly, the binding sites for miR-200c in FN1 mRNA have been experimentally validated[34], suggesting that FN1 could display substantial ceRNA effects. Encouraged by the results of MRE dynamics analyses, we quantified the number of miR-200c and FN1 mRNA in MCF10A cells during TGF-β-induced EMT. Expectedly, the absolute qPCR analyses confirmed that while miR-200c outnumbered FN1 by 4.43-fold in unstimulated MCF10A cells, FN1 quickly outnumbered miR-200c upon TGF-β-induced EMT (Fig. 4a). Importantly, the absolute number of FN1 reached a level similar to that of miR-200c as early as 24 h into EMT and outnumbered miR-200c by 2.77-fold (3035 vs. 1094) at 96 h. Hence, the ceRNA effects through FN1 occurred prior to transcriptional repression of miR-200c by ZEB1, which occurred late into TGF-β-induced EMT because the substantial downregulation of miR-200c was observed late (>120 h) into EMT[26] (Fig. 4a).

To confirm that the ceRNA effect of FN1 could regulate EMT, we first performed computational simulations to estimate the changes of miR-200c binding site occupancies analogous to miR-21. Unlike miR-21, which was upregulated during EMT, miR-200c expression was gradually repressed during EMT in MCF10A cells[35]. More importantly, FN1, the putative miR-200c ceRNA,

displayed a monotonic upregulation (Fig. 4a). Consequently, mathematical simulation showed that the 8mer miR-200c site occupancy gradually decreased as cells undergoing EMT, and declined from about 90% at 0 h to around 55% at 72 h into EMT using MREs estimated by pictar (Fig. 4b). Critically, the reduction in miR-200c binding site occupancy was predominantly due to FN1 because the removal of FN1 MREs from the equation restored the miR-200c binding site occupancy to about 85% at 72 h into EMT (Fig. 4b). Reassuringly, a similar pattern was observed using MREs estimated from targetScan predictions (Supplementary Fig. 7C).

We next performed experimental analyses to demonstrate that FN1 indeed could regulate EMT as a ceRNA. We first knocked down FN1 using siRNA at 48 h into EMT and examined its effects 48 h later. The time point to introduce siRNA was carefully selected to isolate the ceRNA effects from transcriptional repression, which manifested its effects at approximately 120 h, and to target the time point with maximal ceRNA potency, which occurred at 96 h as shown by the dynamics of binding site occupancy (Fig. 4b). Consistent with the postulated regulatory role of FN1 as a ceRNA for miR-200c, the knockdown of FN1 significantly enhanced the expression of CDH1 and miR-200c and repressed the expression of ZEB1 and CDH2 (Fig. 4c, d). Importantly, rescue experiments with FN1 3′UTR containing miR-200c binding sites, but not control 3′UTR lacking the miR-200c sites, restored the expression of the tested genes; this result confirmed that the observed changes indeed reflected ceRNA effects (Fig. 4c, d). We next performed two additional sets of experiments analogous to TGFBI. In the first approach, we depleted DICER1 from MCF10A with DICER1-specific siRNA (Supplementary Fig. 8), and remeasured the impact of FN1 on the core EMT molecules. Consistent with the essential role of mature miRNAs in mediating ceRNA effects, FN1 knockdown had no detectable impact on ZEB1, CDH1, CDH2, and miR-200c in DICER1-less MCF10A cells (Fig. 4e, f). In a second approach, we mutated the seed region of miR-200c binding site in the 3′UTR of FN1 with CRISPR/Cas9 technology (Supplementary Fig. 9), and reexamined the ceRNA activity of FN1 in the absence of endogenous functional miR-200c binding sites. Although pictar predicted two miR-200c binding sites in the 3′UTR of FN1, only one was also predicted by targetScan as "conserved". A careful examination of the secondary structures suggested that only the conserved site is functional because the poorly conserved site is located in a stem region of the modeled secondary structures (Supplementary Fig. 9A, B). We then knocked out the poorly conserved site with CRISPR/Cas9 technology and observed no detectable impact on the ceRNA activities of FN1 (supplementary Fig. 9C–E), confirming that the poorly conserved miR-200c binding site is not functional. We then mutated the conserved miR-200c binding site in FN1 3′UTR with CRISPR/Cas9 and measured the impact on the ceRNA activities of FN1 (Fig. 4g, h, and Supplementary Fig. 9F, G). Reassuringly, FN1 knockdown in MCF10A cells without endogenous functional miR-200c binding site in FN1 3′UTR did not produce significant impact on the key EMT molecules (Fig. 4g, h). In sharp contrast to DICER1-less MCF10A cells, where the addition of exogenous FN1 3′UTR with functional miR-200c binding site displayed no detectable impact on key EMT molecules due to the lack of mature miRNAs (Fig. 4e, f), the introduction of exogenous FN1 3′UTR into MCF10A cells without endogenous functional miR-200c binding site in FN1 3′UTR did generate a small but significant upregulation of ZEB1 and CDH2, further demonstrating that the observed ceRNA activity of FN1 is highly specific and dependent on the presence of mature miRNAs and functional miR-200c binding site in FN1. Finally, FN1 knockdown only impacted the expression levels of genes with experimentally

confirmed miR-200c binding sites, further confirming the specificity of the ceRNA effects associated with FN1 (supplementary Fig. 10A, B).

FN1 is a marker for mesenchymal cells, and its upregulation has been universally associated with EMT[22]. Thus, the key TFs regulating EMT, such as SNAIL and ZEB, were suggested to regulate FN1 expression. Consistent with this hypothesis, the knockdown of SNAI1 or ZEB1 in MCF10A cells significantly reduced FN1 expression (Supplementary Fig. 10C, D). Hence, the key TFs controlling EMT directly upregulate FN1, the ceRNA that dilutes the inhibitory effects of miR-200c prior to the transcriptional repression of miR-200c via ZEB1; this finding further demonstrates that ceRNA is tightly coupled with double-negative feedback loops and is an intrinsic component of the EMT-regulatory circuits.

Unlike A549 cells, in which the key EMT-regulatory miRNA (miR-21) is the most abundant miRNA, miR-200c is only the 15th highest expressing miRNA in MCF10A cells (Supplementary Table 5). Importantly, the higher ranked miRNAs in MCF10A cells are expressed at substantially high levels than miR-200c. For example, the normalized read count for the highest expressing miR-378a-3p is about 28 times higher than that of miR-200c (5,886,078 vs. 211,178). Assuming a linear relationship between normalized read counts and absolute molecule numbers, we estimated that there are about 38,910 miR-378a-3p molecules in MCF10A cells, which is about 12 times higher than FN1 mRNAs at 96 h into EMT. This discrepancy suggested that if the higher expressing miRNAs also target ZEB1, then the ceRNA effect from FN1 is unlikely to be functional. To address this issue, we compared the miRNAs predicted by targetScan to regulate ZEB1 with the top 14 highest expressing miRNAs in MCF10A cells. Reassuringly, none of the 14 miRNAs targets ZEB1 (Supplementary Table 6). This analysis demonstrated that the FN1-miR-200c axis operates independent of the higher expressing miRNAs in MCF10A cells, and is functional despite the presence of excessive miRNAs.

**Mathematical simulation of the EMT-regulatory circuit**. To further demonstrate the role of ceRNA in regulating EMT, we investigated whether a mathematical model incorporating the ceRNA effect of FN1 could capture the dynamics of EMT, which are characterized by the slow dynamics (long half-life) of miR-200s and the fast dynamics of ZEB1, CDH1, and CDH2[28]. We compared the simulation results of the canonical model and the new model that included FN1, the ceRNA for miR-200c (Fig. 5a). To demonstrate that the canonical model could not capture EMT dynamics using miR-200c-related parameters that were consistent with the long half-life of miRNAs, we performed simulations using the canonical EMT model but with modified parameters. Specifically, the parameters were modified such that the majority (90%) of miR-200c in the mRNA–miRNA complexes was recycled, which is consistent with experimental observations. Bifurcation analyses demonstrated that the cells did not reach the mesenchymal state because ZEB1 could not overcome the repression of miR-200c (Fig. 5b, c, Supplementary Fig. 11A, B). We subsequently modified the model to incorporate ODEs describing the kinetics of FN1-related reactions (Supplementary Model 1). Expectedly, simulation using the model incorporating FN1 successfully recaptured the temporal dynamics of key EMT molecules (Supplementary Fig. 11C). Moreover, the bifurcation analysis recapitulated three stable states, suggesting that the incorporation of FN1 successfully captured the dynamic state transitions during EMT (Fig. 5d). Finally, we simulated FN1 knockdown by increasing the degradation rate constant of free

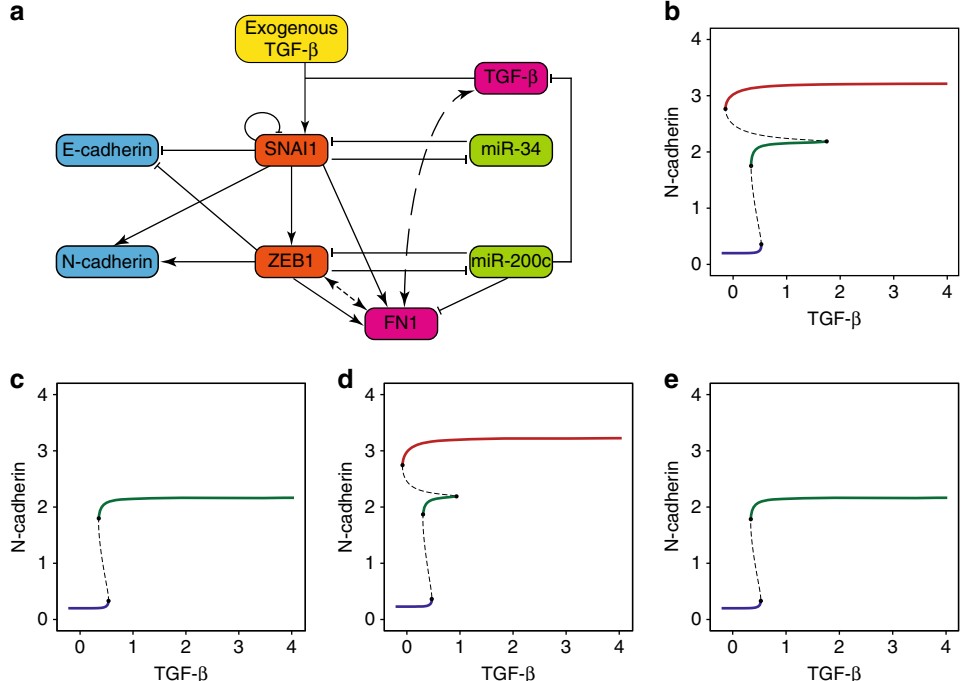

**Fig. 5** Model incorporating FN1 recaptures EMT dynamics. **a** Graph illustrating the EMT-regulatory circuit incorporating FN1. Dotted lines represent ceRNA interactions. **b** Bifurcation analysis using the original EMT model of Zhang et al.[26]. **c** Bifurcation analysis using the original EMT model of Zhang et al. with an increased miR-200c half-life. Specifically, the parameters controlling the recycling rates of miR-200c in the miR-200c-ZEB1 complexes increased from 0.5 to 0.9. **d** Bifurcation analysis using the EMT model incorporating FN1. **e** Bifurcation analysis simulating FN1 knockdown. Specifically, the FN1 degradation rate constant in the EMT model incorporating FN1 increased to 1

FN1 mRNA and observed that miR-200c remained high during the simulation and that the cells failed to enter the mesenchymal state (Fig. 5e, Supplementary Fig. 11D). Reassuringly, the results from simulated FN1 knockdown are consistent with experimental observations following FN1 knockdown (Fig. 4c, d). Taken together, the simulation results confirmed that ceRNA represents an intrinsic component of the EMT-regulatory circuit.

**The abundance of ceRNA determines the reversibility of EMT**. Although ceRNA is an intrinsic component of EMT-regulatory circuits in the two cell lines examined, the dynamics of ceRNA displayed a striking difference. In benign MCF10A cells, FN1 was upregulated and consistently outnumbered miR-200c (Fig. 4a). However, TGFBI was first upregulated and subsequently downregulated in cancerous A549 cells, resulting in a similar number of TGFBI and miR-21 molecules in the mesenchymal state (Supplementary Fig. 4B). Because the stoichiometry between ZEB and miR-200c play critical roles in controlling the reversibility of EMT[36], we hypothesized that the stoichiometry between ceRNA and miRNA could play similar roles.

To examine this hypothesis, we first examined whether A549 cells induced by TGF-β to undergo EMT could also undergo a spontaneous mesenchymal-to-epithelial (MET) transition transition upon the removal of TGF-β. Upon TGF-β treatment, A549 cells underwent EMT and transited from an epithelial phenotype (Fig. 6a) to a mesenchymal phenotype (Fig. 6b). Interestingly, the removal of TGF-β triggered a spontaneous MET and A549 cells reverted back to epithelial phenotype in the presence of control siRNA (Fig. 6c) or control 3′UTR (Fig. 6d). Critically, the reversal to epithelial phenotype upon TGF-β remover was completely blocked with TGFBI 3′UTR (Fig. 6e) or antagomiR-21 (Fig. 6f), suggesting that the high level of miR-21 drives the spontaneous

MET in A549 cells. Reassuringly, the changes in morphology were accompanied by the downregulation of VIM and upregulation of CDH1, which were confirmed through flow cytometry analyses (Fig. 6g, Supplementary Fig. 12A).

We next examined whether MCF10A cells induced by TGF-β to undergo EMT could also undergo a spontaneous MET transition upon the removal of TGF-β. As expected, MCF10A cells transited from an epithelial phenotype (Fig. 7a) to a mesenchymal phenotype (Fig. 7b) upon TGF-β treatment. However, the removal of TGF-β failed to trigger a spontaneous MET and MCF10A cells maintained the mesenchymal phenotype (Fig. 7c). Importantly, FN1 knockdown with specific siRNA did successfully induce MET after TGF-β removal (Fig. 7d), and the effects of siFN1 could be effectively reverse by the addition of FN1 3′UTR (Fig. 7e). Finally, overexpressing miR-200c after TGF-β removal also promoted a transition to the epithelial phenotype, suggesting that both ceRNA and miRNA possess the ability to regulate MET (Fig. 7f). Similar to A549 cells, flow cytometry analyses confirmed that the downregulation of VIM and upregulation of CDH1 occurred simultaneously with morphology transitions in MCF10A cells (Fig. 7g, Supplementary Fig. 12B). Taken together, these data suggested that the stoichiometry between ceRNA and miRNA represents a critical parameter controlling the reversibility of the EMT, which could be a key barrier modulating the efficiency of metastasis.

## Discussion
Here, we showed that a single mRNA dynamically induced during EMT could represent the vast majority of induced MREs and regulate EMT through a ceRNA effect. Because highly expressed mRNAs can readily exceed 1000 copies/cell and because some cancer cells express a substantially lower number of miRNAs than normal tissues[15], these results indicate that, at least

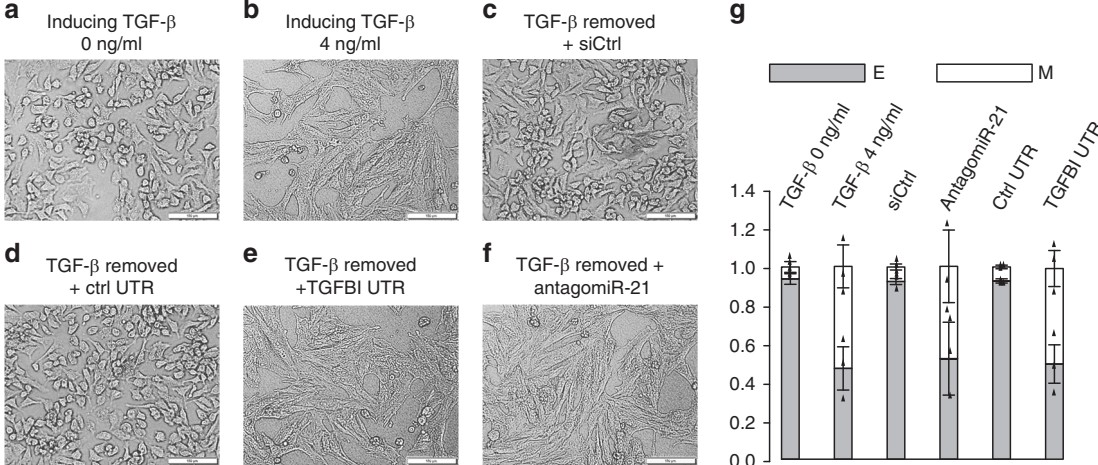

**Fig. 6** CeRNA abundance determines EMT reversibility in A549 cells. **a** Morphologies of untreated A549 cells. **b** Morphologies of A549 cells treated with 4 ng/ml of TGF-β for 7 days to induce EMT. **c** Morphologies of cells treated by 4 ng/ml TGF-β for 7 days followed by TGF-β removal and treated with control siRNA for 3 days. **d** Morphologies of cells treated by 4 ng/ml TGF-β for 7 days followed by TGF-β removal and treated with control UTR for 3 days. **e** Morphologies of cells treated by 4 ng/ml TGF-β for 7 days followed by TGF-β removal and treated with TGFBI UTR for 3 days. **f** Morphologies of cells treated by 4 ng/ml TGF-β for 7 days followed by TGF-β removal and treated with antagomiR-21 for 3 days. **g** Cells in (**a–f**) were subjected to flow cytometry analysis by double staining with CDH1 and VIM antibodies and quantified into E (epithelial cells, gray bars) and M (mesenchymal cells, white bars). The images in (**a–f**) are representative of three independent biological replicates. Scale bars: 100 μm. $n = 3$; error bars indicate the means ± s.d.

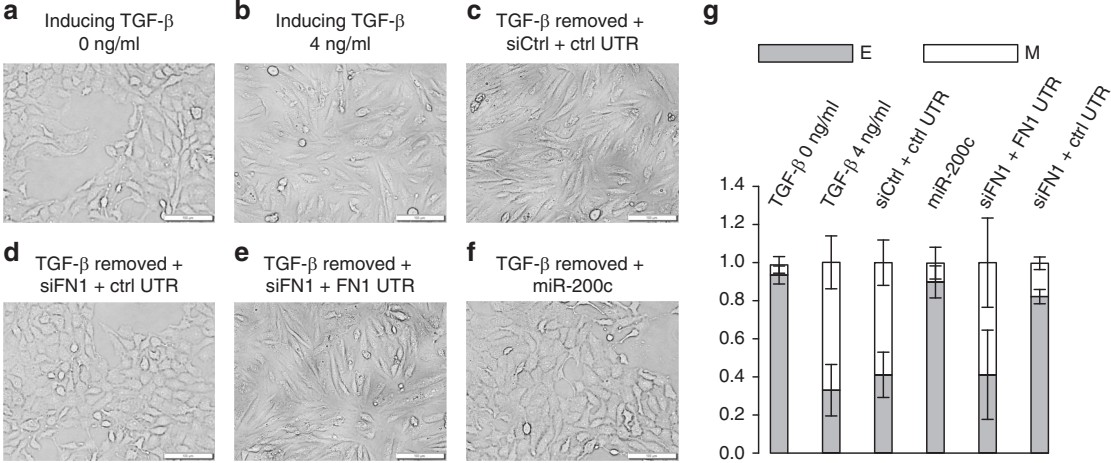

**Fig. 7** CeRNA abundance determines EMT reversibility in MCF10A cells. **a** Morphologies of untreated MCF10A cells. **b** Morphologies of MCF10A cells treated with 4 ng/ml of TGF-β for 7 days to induce EMT. **c** Morphologies of cells treated by 4 ng/ml TGF-β for 7 days followed by TGF-β removal and treated with control siRNA and control UTR for 3 days. **d** Morphologies of MCF10A cells treated by 4 ng/ml TGF-β for 7 days followed by TGF-β removal and treated with siFN1 and control UTR for 3 days. **e** Morphologies of cells treated by 4 ng/ml TGF-β for 7 days followed by TGF-β removal and treated with siFN1 and FN1 UTR for 3 days. **f** Morphologies of cells treated by 4 ng/ml TGF-β for 7 days followed by TGF-β removal and treated with miR-200c for 3 days. **g** Cells in (**a–f**) were subjected to flow cytometry analysis by double staining with CDH1 and VIM antibodies and quantified into E (epithelial cells, gray bars) and M (mesenchymal cells, white bars). The images in (**a–f**) are representative of three independent biological replicates. Scale bars: 100 μm. $n = 3$; error bars indicate the means ± s.d.

in cancer cells that miRNAs are downregulated, the dynamic expression of a single RNA could effectively function as a ceRNA.

Several lncRNAs that are aberrantly overexpressed in metastatic cancers have been reported to regulate EMT via a ceRNA mechanism[6,37–40]. However, these ceRNAs represent a static overexpression that modifies the steady state equilibrium of the mesenchymal state. Recently, Title et al.[25] reported that the miR-200-ZEB1 axis, which is crucial for tumorigenesis, demonstrates extreme signal sensitivity for regulating EMT. In line with their finding, we showed that the dynamically induced ceRNAs are directly coupled with the double-negative feedback loops of EMT, and operate in a dynamic and hypersensitive fashion. These

results suggest that ceRNAs may represent candidate regulators in pathways utilizing TF-miRNA feedback loops.

Previous studies have suggested that the balance between ZEB1 and miR-200 controls the reversibility of EMT[23,36]. However, those analyses were conducted by artificially manipulating the levels of ZEB1 or miR-200; moreover, the mutually inhibitory nature of ZEB1 and miR-200 and the model simulation suggest that the transition into mesenchymal state is largely irreversible[26]. Here, we showed that the stoichiometry between ceRNA and miRNA represents a critical parameter that determines the reversibility of EMT in cancer cell models. Importantly, the cancerous A549 cells express similar numbers of ceRNAs and

miRNAs in the mesenchymal state, and removing TGF-β readily reversed the EMT process. This result suggests that cancer cells may have acquired a distinct mechanism that lowers the energy barrier of the EMT to MET transition by dynamically modulating ceRNA levels. Hence, the stoichiometry between miRNA and ceRNA may represent a novel parameter to intervene metastasis.

## Methods

**Cell culture**. A549 cells and MCF10A cells were obtained from cell bank of Chinese Academy of Sciences and have been test to be free of mycoplasma contamination. The identities of cells have been confirmed by STR profiling. A549 cells were maintained in Dulbecco's Modified Eagle's Medium:Nutrient Mixture F-12 (DMEM/F-12) (Corning) supplemented with 10% fetal bovine serum (FBS). MCF-10A cells were maintained in Dulbecco's Modified Eagle's Medium:Nutrient Mixture F-12 (DMEM/F-12) (Corning) supplemented with 5% Horse Serum (Invitrogen #16050-122), 20 ng/ml EGF, 0.5 mg/ml Hydrocortisone, 100 ng/ml Cholera Toxin, 10 µg/ml Insulin, and 1% Pen/Strep (100× solution, Invitrogen #15070-063). Importantly, numerous studies have shown that there is substantial amount of TGF-β1 in FBS, which is sufficient to induce partial EMT[26,41]. Moreover, MCF10A cells could undergo spontaneous EMT when cultured at low confluence[42,43]. To successfully induce EMT, cells were seeded in serum-free medium for 24 h to reach ∼70% confluence before subject to TGF-β1 treatment. TGF-β1 (Gibco® PHG9214) was added to a final concentration of 4 ng/ml and medium with TGF-β1 was replaced every other day.

**RNA isolation and qRT-PCR**. Total RNA was isolated from cells using TRIzol reagent (TaKaRa). Reverse transcription was performed using the PrimeScript™ RT Reagent Kit (TaKaRa) and gDNA Eraser (Perfect Real Time) according to the manufacturers' instructions. qRT-PCR was performed using the SYBR Premix Ex Taq (Tli RNase H Plus) system on an ABI Step One Plus machine (Applied Biosystems). The experiments were performed in triplicate, and the values of mRNAs were normalized to that of GAPDH and the values of miRNAs were normalized to that of U6 snRNA. The primer sequences were as follows: U6 forward, CTCGCTTCGGCAGCACA; U6 reverse, AACGCTTCACGAATTTGCGT; Unified miRNA reverse primer, CCAGTGCAGGGTCCGAGGTA; miR-21-5p, GCCCGCTAGCTTATCAGACTGATG; miR-200c-3p, GCCCGCTAATACTGCCGGGTAAT; miR-21-5p-RT, GTCGTATCCAGTGCAGGGTCCGAGGTATTCG-CACTGGATACGACTCAACA;

miR-200c-3p-RT, GTCGTATCCAGTGCAGGGTCCGAGGTATTCGCACTGGATACGACTCCATC; DICER1 forward, GAGCTGTCCTATCAGATCAGGG; reverse, ACTTGTTGAGCAACCTGGTTT; FOXP1 forward,TGGCATCTCATAAACCATCAGC; FOXP1 reverse, GGTCCACTCATCTTCGTCTCAG; TGFBI forward, CTTCGCCCCTAGCAACGAG, TGFBI reverse, TGAGGGTCATGCCGTGTTT; CDH1 forward, TCAGGCGTCTGTAGAGGCTT; CDH1 reverse, ATGCACATCCTTCGATAAGACTG; CDH2 forward, ACAGTGGCCACCTACAAAGG; CDH2 reverse, CCGAGATGGGGTTGATAATG; FN1 forward, GGAGTTTCCTGAGGGTTT; FN1 reverse, GCAGAAGTGTTTGGGTGA;

ZEB1 forward, AGTGGTCATGATGAAAATGGAAC; ZEB1 reverse, AGGTGTAACTGCACAGGGAGC; ETS2 forward, CCCCTGTGGCTAACAGTTACA; ETS2 reverse, AGGTAGCTTTTAAGGCTTGACTC; JUNB forward, ACAAACTCCTGAAACCGAGCC; JUNB reverse, CGAGCCCTGACCAGAAAAGTA; HNF4A forward, CACGGGCAAACACTACGGT; HNF4A reverse, TTGACCTTCGAGTGCTGATCC; TGFBI 3′UTR forward, CAGCCTCATGGGAAGTCCT; TGFBI 3′UTR reverse, ACATTTGACAGAACATTTCAACTCA; FN1 3′UTR forward, TGCTTCGAAGTATTCAATACCG; FN1 3′UTR reverse, TATAAAAATACTGGGAAAAATTGATAAA; SNAIL1 forward, TCGGAAGCCTAACTACAGCGA; SNAIL1 reverse, AGATGAGCATTGGCAGCGAG; GAPDH forward, GGAGCGAGATCCCTCCAAAAT; GAPDH reverse, GGCTGTTGTCATACTTCTCATGG.

**Absolute quantification of mRNAs and miRNAs**. For absolute qPCR, sequence fragments of TGFBI mRNA (hg19, chr5:135,398,916–135,399,507) and FN1 mRNA (hg19, chr2:216,226,278–216,300,525) were synthesized by Sangon Biotech (Shanghai) Co., Ltd. and used to generate standard curves using the method described by Dhanasekaran et al.[44]. For the absolute qPCR of mir21-5p and mir200c-3p, the standard curves were generated using synthetic miRNAs (Shanghai GenePharma Co., Ltd.), and the absolute copy numbers of mature miRNAs were obtained using Hairpin-it miRNA qRT-PCR kits (Shanghai Gene-Pharma Co., Ltd, catalog # E01006 and E22001) according to Gong et al.[45]. The following primer sequences were used: TGFBI forward, TGGGGCTCATAAAA-CATGAA; TGFBI reverse, CCTCCAAGCCACGTGTGTAGAT; FN1 forward, ACCAACCTACGGATGACTCG; FN1 reverse, GCTCATCATCTGGCCATTTT.

**Transient siRNA knockdown and transfection of 3′UTR**. Cells were transfected with a negative control siRNA (Silencer R, Life Technologies) or with siRNAs targeting FOXP1, TGFBI, FN1, ETS2, JUNB, or HNF4α at a final concentration of 100 nM using Lipofectamine 2000 reagent (Invitrogen) or DharmaFECT Formulation 1 (Dharmacon) according to the manufacturer's instructions. Co-

transfection of siRNA and plasmids containing 3′UTR of FN1 or TGFBI were performed using DharmaFECT Duo (Dharmacon) according to the manufacturer's instructions. Data were analyzed with t test after confirming that the control and treatment groups exhibit similar variance. The following are the siRNA sequences: TGFBI-Homo, sense, CUUGCCAACAUCCUGAAAUTT, antisense, AUUUCAGGAUGUUGGCAAGTT; FN1-homo, sense, GUCCUGUCGAAGUAUUUAUTT, antisense, AUAAAUACUUCGACAGGACTT;

SNAI1-Homo, sense, CAGAUGUCAAGAAGUACCATT, antisense, UGGUACUUCUUGACAUCUGTT; FOXP1-Homo 1, sense, GCAGCAACCACUUACUAGATT, antisense, UCUAGUAAGUGGUUGCUGCTT; FOXP1-Homo 2, sense, GCUCAAGGCAUGAUUCCAATT, antisense, UUGGAAUCAUGCCUUGAGCTT; FOXP1-Homo 3, sense, GUACAGCCCAAUGUAGAGUTT, antisense, ACUCUACAUUGGGCUGUACTT; ETS2-homo, sense, GCCUCAAUAAGCCAACCAUTT, antisense, AUGGUUGGCUUAUUGAGGCTT; JUNB-Homo, sense, ACAAGGUGAAGACGCUCAATT, antisense, UUGAGCGUCUUCACCUUGUTT; HNF4A-homo, sense, GCAGCUGCUGGUUCCUCGUUTT, antisense, AACGAGAACCAGCAGCUGCTT; ZEB1-homo, sense, GGAUCAACCACCAAUGGGUUTT, antisense, AACCAUUGGUGGUUGAUCCTT; DICER1-Homo-1008, sense, GGACCAUUUACUGACAGAAUTT, antisense, UUCUGUCAGUAAAUGGUCCTT; DICER1-Homo-2123, sense, GGCCAUUGGACACAUCAAUTT, antisense, AUUGAUGUGUCCAAUGGCCTT;

DICER1-Homo-4380, sense, CCUCCUGGUUAUGUAGUAAUTT, antisense, UUACUACAUAACCAGGAGGTT.

For rescue experiments, the 3′UTR for TGFBI (hg19, chr5:135,399,223-135,399,462) and FN1 (hg19, chr2:216,225,719-216,226,087) were synthesized by Sangon Biotech (Shanghai) Co., Ltd. and cloned in pcDNA3.1 plasmid.

**Mutagenesis with CRISPR-Cas9**. The CRISPR-Cas9 system used in this study is a two-vector system (lentiCas9-Blast, catalog#:52962; and lentiGuide-Puro, Catalog#:52963) and was purchased from Addgene. A549 and MCF-10A were infected by the lentivirus containing LentiCas-Blast first (MOI = 0.3) and selected with 8 µg/ml Blasticidin for 5 days. To generate desired vector for mutagenesis, the guide RNA vector was digested with BsmBI (Fermentas) and a pair of annealed oligos containing the sgRNA was cloned into the single guide RNA scaffold. The A549-Cas9 cells and MCF10A-Cas9 cells were infected with the lentivirus contained guide RNA (MOI = 0.3) and selected with 1ug/ml Puromycin for 3 days. To generate single clone for downstream experiments, the pool of infected cells were selected using the 96-well plate limiting dilution assay. The knockout efficiency was evaluated by DNA sequencing and TA clone sequencing. The following are the sgRNA sequences:

DICER1, AAAGAAAGGACCCATTGGTG; FN1 conserved miR-200c binding site, CTCAGTATTTTAAATGAAGT; FN1 poorly conserved miR-200c binding site, GTATTTTTATACGGAAAAAA; FOXP1 miR-21 binding site, CTCCCATCCACTCATAAGCT; TGFBI miR-21 binding site, CCCTTGCACAGCTGGAGAAA.

**Transwell migration and invasion assay**. The in vitro cell migration assay was performed using Transwell chambers (8 µm pore size; Costar). Cells were plated in serum-free medium ($2 \times 10^4$ cells per Transwell). Medium containing 10% FBS in the lower chamber served as a chemoattractant. After 24 h, the nonmigrating cells were removed from the upper face of the filters using cotton swabs and the migratory cells located on the lower side of the chamber were stained with crystal violet, air dried, photographed and counted. Images of six random fields at ×10 magnification were captured from each membrane, and the number of migratory cells was counted. Similar inserts coated with Matrigel were used to determine cell's invasive potential in the invasion assay. Data were as analyzed with t test after confirming that the control and treatment groups exhibit similar variance.

**Fluorescence microscopy and staining for CDH1 and VIM**. For CDH1 staining, $2 \times 10^5$ cells were seeded onto a 30-mm confocal dish, fixed in 4% paraformaldehyde, blocked in 10% goat serum in PBS, and probed with specific primary antibodies for E-cadherin (Invitrogen, A15757, 1:100). For VIM staining, the cells were fixed in 4% paraformaldehyde, permeabilized in 0.2% Triton X-100, blocked in 10% goat serum in PBS, and probed with specific primary antibodies (Abcam, ab92547, 1:100) for 1 h. To detect the nuclei, the cells were co-stained with 4–6-diamidino-2-phenylindole (DAPI; Invitrogen). The cells were observed on a Leica TCS SP8 Confocal Laser Scanning Microscope. The images were analyzed using Zeiss ZEN software.

**Immunoblotting analyses**. Protein lysates were prepared in the presence of PIC (protease–inhibitor complex) and PMSF. Twenty microgram aliquots were separated on 8% sodium dodecyl sulphate polyacrylamide electrophoresis gels, and the proteins were transferred onto a polyvinylidene difluoride membrane (Merck Millipore). The membrane was incubated for 1 h in blocking buffer (tris-buffered saline containing 0.1% Tween (TBS-T), and 5% nonfat dry milk) followed by incubation overnight at 4 °C with the primary antibodies for DICER1 (CST, D38E7, 1:1000), TGFBI (Abcam, ab170874, 1:2000), FOXP1 (CST, 4402, 1:1000), FN1 (Santa Cruz, sc-9068, 1:200), CDH1 (CST, 3195s, 1:1000), and CDH2 (CST, 14215s, 1:1000). After washing with TBS-T, the blot was incubated with

horseradish peroxidase (HRP)-conjugated secondary antibody or IRDye800-conjugated secondary antibody and the signals were visualized using an enhanced chemiluminescence system according to the manufacturer's instructions (Kodak) or Odyssey (LI-COR). The uncropped and unprocessed scans of the most important blots are provided in the Source Data file.

**Luciferase reporter assay**. The cells were seeded onto six-well plates at a density of $1 \times 10^5$ cells/well and cultured until reaching 40% confluence. The cells were subsequently transfected with 3′UTR of FOXP1 or a negative control 3′UTR lacking the seed region of the miR-21 binding site (3′UTRs were cloned into the pMIR-Reporter Luciferase vector) at a final concentration of 1 μg/well using Lipofectamine 2000 reagent (Invitrogen) according to the manufacturer's instructions. Forty-eight hours after transfection, the cells were seeded onto 96-well plates at a density of $1 \times 10^4$ cells/well, and the luciferase signal was detected after 24 h using a luciferase assay system (Promega, E2510) according to the manufacturer's instructions.

**Flow cytometry**. The cells were fixed with 4% paraformaldehyde (15 min at room temperature). For VIM staining, the cells were permeabilized by slowly adding ice-cold 100% methanol to pre-chilled cells under gentle vortexing to a final concentration of 90%. To block nonspecific binding, the cells were suspended in incubation buffer (10% goat serum in PBS, Gibco®) and incubated on ice for 30 min. Staining was performed using CDH1 (Life Technologies, A15757, 1:100) or VIM (Abcam, ab203428, 1:100) antibodies conjugated to fluorescent dye at room temperature for 1 h. The cells were analyzed using a Beckman Coulter CytoFLEX S system, and the data analysis was performed using CytExpert_1.2.11.0 software.

**Estimating number of MREs using gene-expression data**. We estimated the relative number of MREs by multiplying the transcripts per million (TPM) of each transcriptional isoform containing corresponding miRNA binding site with the number of conserved miRNA binding sites in their 3′UTR as predicted by targetScan or pictar[13,46]. Specifically, for miR-21 we used the time course gene-expression data of TGF-β-induced EMT in A549 cells (GSE69667)[22]. For miR-200c, we used the time course gene expression data of MCF10A cells over-expressing SNAI1 (GSE52592)[21]. Because the MCF10A time course data is measured by microarray, we first estimated the gene expression TPM in unstimulated MCF10A cells (GSE48213)[47], and extrapolated the gene expression TPM during EMT by multiplying the base TPM with gene expression fold changes estimated from microarray time course data in SNAI1 overexpressing MCF10A cells. Finally, assuming absolute qPCR provides a more accurate measurement of the number of molecules in cells, we scaled number of MREs derived with RNASEQ TPM by the ratio between qPCR measured number of TGFBI or FN1 with their corresponding TPM, and used the scaled MREs for downstream analysis.

**MicroRNA sequencing and data analysis**. The sequencing libraries were constructed according to the protocol for the Illumina small RNA Sample preparation kit. Sequencing was performed on the Illumina HiSeq 2000 sequencer. Library construction and sequencing were performed at the Genergy Biotech Co., Ltd. (Shanghai). MiRNA expression was analyzed by miRdeep2.0.0.7[48] and differentially expressed microRNAs were identified using an FDR cutoff value of 0.05 and absolute log2-fold change greater than 1.

**Mathematical modeling of miRNA binding site occupancy**. We utilized the model developed by Jens and Rajewsky[49] to simulate the changes of miRNA binding site occupancy during EMT. We first estimated the number of MREs using the conserved miRNA binding sites predicted by targetScan 7.2 and the miRNA binding sites predicted by pictar (https://pictar.mdc-berlin.de/). The number of miRNA molecules were determined by absolute qPCR. The software developed by Jens et al. was downloaded from http://dorina.mdc-berlin.de/public/rajewsky/rna_competition/, and the "simplified model" was used to estimate the miRNA binding site occupancies.

**Mathematical modeling of EMT-regulatory circuits**. We utilized the Cascading Bistable Switches (CBS) model proposed by Zhang et al.[26] for the mathematical simulation. We generated the following modifications to incorporate FN1 into the CBS model. First, we assumed that the majority (90%) of miR-200c could be recycled from miR-200c-ZEB1 complexes, resulting in a miR-200c half-life that was consistent with experimental observations. Specifically, we changed the recycling ratio λ (1–5) in the ZEB1/miR-200 module from 0.5 to 0.9. Second, we added ODEs describing the kinetics of FN1 mRNA and estimated the related parameters using experimental gene expression data. Specifically, we modeled one conserved binding site of miR-200c in FN1 based on targetScan prediction and employed Hill functions to model the transcriptional regulation of FN1 by SNAI1 and ZEB1. One-parameter bifurcation analyses and 300-h time course analyses were performed using Oscill8. The parameters that differed from those of the CBS model of Zhang et al. are provided in Supplementary Table 7.

## Data availability

The miRNA-seq data reported in this study have been deposited in GEO under accession number GSE87358. All other data are available from the authors upon reasonable request.

## Code availability

The ODEs describing the EMT model with FN1 are provided in Supplemental Note.

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

## Acknowledgements

The authors would like to thank the Cell Sorting Core at Shanghai Institute for Advanced Immunochemical Studies (SIAIS) in ShanghaiTech University for assistance with FACS analyses. This work was supported in part by National Key R&D Program of China (No. 2017YFA0505500), the Strategic Priority Research Program of the Chinese Academy of Sciences (No. XDB13040700), and the National Natural Science Foundation of China (31271413, 31671380, and 31771476).

## Author contributions

Y.L. and S.D. performed the experiments. M.X., W.F. and K.Z. performed the computational analysis and bifurcation analyses. L.Z., H.L., G.J. and L.W. contributed to the experiments. X.H., L.C. and P.W. conceived the project and supervised the study. P.W. wrote the paper with inputs from all authors. All authors approved the final paper.

## Additional information

**Competing interests:** The authors declare no competing interests.

