## [Peer Review File · Nature Communications]

Reviewers' Comments:

Reviewer #1 (Remarks to the Author)

The authors describe ceRNA effects of specific genes during the induction of EMT, which is an important process in tumor biology. The subject is overall of interest. The paper is well written and data presented very clearly. The first parts, pertaining to the roles of FOXP1 in EMT and the regulation of FOXP1 by miR-21, as well as the parts where the authors study transcriptional regulation of TGFBI and FN1, are solid and rigorous. The main claims in the paper and most of the novelty pertain to the ceRNA hypothesis, and they are substantially more questionable, as described below.

Major comments:

1) The authors describe potential ceRNA activity taking place during EMT, driven by high expression of single target genes– TGFBI for mir-21 and FN1 for mi-200c. In both cases ceRNA activity is reported to be driven by a single strong target site. The authors claim that these effects are plausible because the target abundance of these genes approaches the miRNA abundance, though recent computational analysis and data (reviewed in Jens and Rajewsky Nature Reviews Genetics 2015) suggest that even such levels for a single mRNA carrying a single target site are unlikely to be consequential, as the number of other binding sites for the miRNA in the transcriptome and overall target occupancy need to be considered. Jens and Rajewsky also provide specific criteria which need to be met for the ceRNA effect to be likely. The paper by Boson et al. from the Sharp lab that the author cite (ref. 11) also concludes that competition is unlikely when the miRNA levels are high, as they are for mir-21 and mir-200c (>1,000 copies per cell, according to the authors). In the discussion, the authors claim that “A central debate concerning the physiological relevance of ceRNA is whether the expression level of ceRNAs could approach the abundance of miRNAs regulating a critical biological process.”, but actually the debate is more about whether a single gene, no matter how abundant, with a single target site, can compete against the pool of thousands of sites found in the rest of the expressed transcriptome. Do the changes observed by the authors meet the criteria set forth in the published quantitative models of ceRNA function from the Rajewsky, Bartel and Sharp labs? That seems unlikely, and if they do, it should be specifically explained and discussed

2) As the ceRNA effects of individual mRNAs with a single binding site are not plausible, the authors must very convincingly show that direct competition is indeed happening in their setting. Currently, this is shown using individual reporter assays, but these may suffer from indirect effects. Further, the authors used siRNAs for knockdown of TGFBI and FN1, which might saturate the RISC machinery, thus indirectly affecting microRNA efficacy. To address this, the authors need to: a) show using RNA-seq or microarray analysis that there is specific de-repression of target genes containing seed sites for miR-21 and miR-200c in the two systems when TGFBI and FN1 are perturbed (this is best done using Sylamer analysis); b) show what happens when the ceRNAs are targeted with a different method – ideally, by mutating the specific target sites in TGFBI and FN1 using CRISPR (as suggested by Jen and Rajewsky in their review) or using CRISPRi for target silencing, which would not suffer from potential RISC overloading effect; c) show that the effects of TGFBI on FOXP1 and of FN1 on ZEB1 are dependent on the microRNA pathway (e.g., by deleting Dicer and showing the effect is abrogated).

3) The ceRNA effect is supposed to happen by competition for miRNA binding, but regular sites are not supposed to reduce the expression level of the miRNA. How do the authors explain the increase in miR-21 levels when using siTGFBI in Fig. 3C, which is abrogated when the TGFBI 3'UTR is used? Especially since mir-21 is not reduced when TGFBI is induced naturally during EMT (Fig. 3B)? Similarly, how is the induction in miR-200c levels in Fig. 4B explained?

Minor comments:

4) The authors claim that cancer cells “typically express lower levels of miRNAs” (Page 4), but as

far as I know that are reports for both decreased and increased overall miRNA abundance in tumors. Is there evidence that the overall miRNA abundance in the cells studied is indeed low compared to hepatocytes or other cell types where the ceRNA hypothesis was studied before? This should be cited or shown.

5) Which isoform of FN1 is induced? According to TargetScan there are two isoforms and only one of them is targeted by mir-200.

Reviewer #2 (Remarks to the Author)

In this paper, the authors investigate the importance of ceRNA regulation within the context of EMT. To do so, they characterize two specific examples of ceRNA-miRNA pairs in two different cell lines, and conclude that ceRNA regulation occurs and plays an important role in EMT. Furthermore, evaluation of miRNA and ceRNA stoichiometry provides some support for ceRNA in this specific cancer cell line context.

Comments:

Major:

1. The authors provide some evidence pointing to a relevant ceRNA mediated mechanism modulating the dynamics of EMT. However, like numerous previous studies they fall short of proving this genetically. The authors should capitalize on their cell system and mutate the miR-21 and miR-200c binding sites (by single point mutations) in the TGFBI and FN1 genes, respectively, using readily available genome editing approaches and test if this affects EMT. These are key experiments that are essential for the conclusions of the manuscript.
2. The authors, throughout the manuscript, measure EMT markers and study cell lines in surrogate assays, without determining of whether the postulated ceRNA-miRNA-target gene networks are relevant in in vivo conditions (i.e. metastasizing and non-metastasizing mouse or human tumors).
3. A recent study by Denzler et al. (Mol Cell. 64: 565–579, 2016) found that cooperative binding of proximal sites for the same or different miRNAs can increase potency and therefore make ceRNA effects more likely. If such closely-spaced sites exist should be investigated and discussed.
4. In lines 104-116, the authors state that there is a discrepancy between a long miR-200c half-life and rapid ZEB1/CDH1/CDH2 dynamics, concluding that unknown mechanisms must modulate miR-200c expression in order to resolve this issue. This view seems reductionist and ignores factors other than miR-200c that may regulate ZEB1 and its downstream effectors, such as SNAIL1 (Dave et al., 2011, JBC).
5. There is some contradiction related to the point at which FOXP1 reaches equilibrium – is it 48h (line 157) or 96h (line 168)? This is relevant to evaluating the dynamics of the system.
6. Does miR-21 regulate other EMT genes? The authors state that miR-21 knockdown leads to enhanced EMT (lines 198-199), but there is no experimental evidence that this is occurring through FOXP1 repression. In order to make this claim, it should be experimentally shown that mutation of the FOXP1 miR-21 binding site prevents enhanced migration/invasion upon miR-21 knockdown.
7. There is no explanation as to why different cell lines are used to study the miR-21-TGFBI and miR-200c-FN1 regulatory networks. Expression levels of both networks in both cell lines would be informative to help understand stoichiometric requirements for such ceRNA events to occur, or explain why they do not occur under specific circumstances. Furthermore, the authors should discuss the evidence that the observed expression changes/dynamics are relevant for certain primary (as opposed to cell lines) cancer types.

Minor:

1. Lines 292-301 should be rewritten more clearly, as it is not clear to what “transcriptional repression” refers (presumably to ZEB1 repression of miR-200c?), nor how the time points were selected (how is the point of “maximal ceRNA potency” calculated?).

Reviewer #3 (Remarks to the Author)

The manuscript by Du et al reports that TGFbeta and Fibronectin mRNA can act as ceRNAs in the control of EMT in A549 lung adenocarcinoma cells and MCF10A immortalized mammary cells respectively. The authors show that the level of expression of miR-21 and miR-200C are modulated in the presence of their corresponding ceRNA. The kinetics model initially developed by Tian et al., (2015) has been modified to introduce the contribution of the ceRNA. The findings are of interest to further unravel mechanisms driving epithelial cell plasticity.

Remarks

A 549 adenocarcinoma cells are known to be morphologically very heterogeneous with clusters of epithelial-like cells scattered among cells with a pronounced mesenchymal morphology. MCF10A has a basal-like phenotype as most normal-like mammary epithelial cells and again raises issues in terms of epithelial mesenchymal transition. The authors should at least discuss this issue. The result section describing Figure 6 needs to be considerably improved; Each frame should be described accurately. Clearly the reversibility to an epithelia-like morphology is not obvious.

Reviewers' comments:

Reviewer #1 (Remarks to the Author):

The authors describe ceRNA effects of specific genes during the induction of EMT, which is an important process in tumor biology. The subject is overall of interest. The paper is well written and data presented very clearly. The first parts, pertaining to the roles of FOXP1 in EMT and the regulation of FOXP1 by miR-21, as well as the parts where the authors study transcriptional regulation of TGFBI and NF1, are solid and rigorous. The main claims in the paper and most of the novelty pertain to the ceRNA hypothesis, and they are substantially more questionable, as described below.

Major comments:

1) The authors describe potential ceRNA activity taking place during EMT, driven by high expression of single target genes— TGFBI for mir-21 and FN1 for mi-200c. In both cases ceRNA activity is reported to be driven by a single strong target site. The authors claim that these effects are plausible because the target abundance of these genes approaches the miRNA abundance, though recent computational analysis and data (reviewed in Jens and Rajewsky Nature Reviews Genetics 2015) suggest that even such levels for a single mRNA carrying a single target site are unlikely to be consequential, as the number of other binding sites for the miRNA in the transcriptome and overall target occupancy need to be considered. Jens and Rajewsky also provide specific criteria which need to be met for the ceRNA effect to be likely. The paper by Boson et al. from the Sharp lab that the author cite (ref. 11) also concludes that competition is unlikely when the miRNA levels are high, as they are for mir-21 and mir-200c (>1,000 copies per cell, according to the authors). In the discussion, the authors claim that “A central debate concerning the physiological relevance of ceRNA is whether the expression level of ceRNAs could approach the abundance of miRNAs regulating a critical biological process.”, but actually the debate is more about whether a single gene, no matter how abundant, with a single target site, can compete against the pool of thousands of sites found in the rest of the expressed transcriptome. Do the changes observed by the authors meet the criteria set forth in the published quantitative models of ceRNA function from the Rajewsky, Bartel and Sharp labs? That seems unlikely, and if they do, it should be specifically explained and discussed

Response: We thank reviewer for pointing out this critical issue of our study, which we didn't address with enough details in our original manuscript. As the reviewer points out, the issue is "whether a single gene, no matter how abundant, with a single target site, can compete against the pool of thousands of sites found in the rest of the expressed transcriptome.". We specifically addressed this issue by performing mathematical modeling analyses as suggested by the reviewer using the models developed by the Rajewsky and Sharp labs. The simulation results suggested that a single gene, FN1 in MCF10A and TGFBI in A549, indeed can compete against the pools of other MREs to induce sizable changes in the miRNA binding site occupancy.

Because the computational prediction of miRNA binding sites is known to suffer from high false positive rates (Pinzón et al. *microRNA target prediction programs predict many false positives. Genome Res.* 2017 Feb;27(2):234-245.), we extended our original study by considering predictions from both targetScan and pictar to estimate the number of MREs. We then first adopted the model of Rajewsky lab and calculated miRNA binding site occupancy during EMT using the "simplified model" as described by the authors. The mathematical modeling clearly showed that the dynamic expression change of FN1 or TGFBI during EMT can substantially change the MRE occupancies. Specifically, without TGFBI, the miR-21 site occupancy is consistently over 90% during the entire course of EMT in A549 cells. In sharp contrast, the miR-21 site occupancy in FOXP1 dropped to around 65-70% (targetScan-based MREs) at 12 to 36 hours when the MRE from TGFBI is included (Updated Fig3B). Simulations using pictar-based MREs generate a similar pattern in MRE dynamics, where the inclusion of TGFBI MREs induced a drop of site occupancy at 12 to 36 hours into EMT, albeit of a smaller magnitude (site occupancy dropped from ~97% to 88%, supplementary Fig. 4D). Although the site occupancy dynamics during EMT in MCF10A cells is different, a clear reduction in site occupancy induced by FN1 is also observed. When the MRE from FN1 is excluded, the miR-200c site occupancy declines gradually from around 90% at the start of EMT, but still above 80% at 120 hours into EMT using pictar-based MREs. In sharp contrast, the addition of FN1 MREs dramatically accelerated the reduction of miR-200c site occupancy, and the miR-200c site occupancy dropped from 85% without FN1 to around 55% with FN1 (pictar-based MRE) at 72 hours into EMT (Fig. 4B). A similar albeit smaller effect of FN1 is also observed with targetScan-based MREs, where the site occupancy dropped from ~50% to 35% at 72 hours into EMT (Supplementary Fig. 7C).

We also performed simulations using the model from the Sharp lab, which is mathematically equivalent to the “simplified version” of the Rajewsky model. We implemented a Python routine based on the scipy library to solve the equations based on the Sharp model. As expected, identical changes of site occupancies were obtained despite the different solvers utilized. Hence these data are not included in the updated manuscript.

Although the simulated reduction in site occupancy owing to ceRNA is only up to 30%, representing a mild ceRNA effect, site occupancy changes at the estimated magnitude could indeed have a potent downstream effect owing to the following observations. First, the MREs were estimated by miRNA binding site prediction programs, which are known to have high false positive rates (Pinzón et al. *microRNA target prediction programs predict many false positives*. *Genome Res.* 2017 Feb;27(2):234-245.). However, the miRNA binding sites for FN1 and TGFBI are experimentally validated. Hence, we expect that the MRE from FN1 or TGFBI could represent an even greater portion of the total MREs than current estimation, and consequently, their expression changes could lead to a larger change in site occupancy, and a bigger ceRNA effect. Secondly, and more importantly, the ceRNA interactions are tightly coupled with double negative feedback loops between ZEB-miR-200c and FOXP1-miR-21 in our study. A critical characteristic of double negative feedback loops is that they can generate switch like behavior, where a small change in input leads to dramatic expression changes of coupled molecules, a phenomenon known as hypersensitivity. Thus it is highly likely that the ceRNA signals generated by the site occupancy changes are amplified by the feedback loops, resulting in substantial changes in the downstream molecules during EMT, which is consistent with our experimental observations and mathematical modeling results.

Taken together, these new analyses demonstrated that a single highly induced mRNA can indeed display noticeable ceRNA effects in these EMT models, given that the MREs from other RNAs are of smaller number, and that the ceRNA signals are tightly coupled with the hypersensitive double feedback loops. We have incorporated these modeling results into updated Figure 3, Figure 4, supplementary Figure 4 and supplementary Figure 7.

2) As the ceRNA effects of individual mRNAs with a single binding site are not plausible, the authors must very convincingly show that direct competition is indeed happening in their setting. Currently, this is shown using individual reporter assays, but these may suffer from indirect effects. Further, the authors used siRNAs for knockdown of TGFBI and FN1, which

might saturate the RISC machinery, thus indirectly affecting microRNA efficacy. To address this, the authors need to: a) show using RNA-seq or microarray analysis that there is specific de-repression of target genes containing seed sites for miR-21 and miR-200c in the two systems when TGFBI and FN1 are perturbed (this is best done using Sylamer analysis); b) show what happens when the ceRNAs are targeted with a different method – ideally, by mutating the specific target sites in TGFBI and FN1 using CRISPR (as suggested by Jen and Rajewsky in their review) or using CRISPRi for target silencing, which would not suffer from potential RISC overloading effect; c) show that the effects of TGFBI on FOXP1 and of FN1 on ZEB1 are dependent on the microRNA pathway (e.g., by deleting Dicer and showing the effect is abrogated).

Response: We agree that more convincing evidences are needed to support our conclusion and have performed additional experiments/analyses as suggested by the reviewer.

First, we carried out new experiments to show that there is specific de-repression of target genes containing seed sites for miR-21 and miR-200c in the two systems when TGFBI and FN1 are perturbed. Although Sylamer analysis can provide a systematic answer, it is not suitable in our case. For both miR-21 and miR-200c, the estimated ceRNA effects from our response to comment # 1 is mild, suggesting that the direct gene expression changes induced by ceRNA effects will not be dramatic. RNA-seq or microarray are not robust to detect small differential expressions changes, and consequently, Sylamer analysis probably will not produce reliable results. To combat this issue, we performed more sensitive qPCR analyses on a large set of genes with experimentally validated miR-21 or miR-200c binding sites. To control for indirect effect, we selected genes with experimentally validated binding sites for other high expressing miRNAs (the second highest expressing let-7 in A549, and the highest expressing miR-385 in MCF10A) but lacking binding sites for miR-21 or miR-200c. Consistent with our model, we consistently observed expression changes in genes with MREs for miR-21 or miR-200 when perturbing TGFBI or FN1, and no significant changes in control genes, confirming that the observed ceRNA effects are direct and highly specific. The qPCR results have been included in supplementary Figure 6 and 10.

Second, we have utilized CRISPR/Cas9 technology to mutate the seed region of miR-21 binding site in TGFBI 3'UTR and seed region of the miR-200c binding site in FN1 3'UTR, and repeated the same assays originally performed with siRNAs. Consistent with the

original siRNA data, knocking out binding site in FN1 or MCF10A completely abolishes ceRNA effect, confirming that the observed ceRNA effect is not due to RISC overloading, but require the functions miRNA binding sites in putative ceRNA molecules (updated Fig. 3-4, supplementary Fig. 3, 5-6).

Finally, we also deleted Dicer in A549 (via CRISPR/Cas9) and MCF10A (siRNA based) and confirmed that the ceRNA effects of TGFBI or FN1 are abrogated without mature miRNAs, confirming that the effects of TGFBI on FOXP1 and of FN1 on ZEB1 are dependent on the microRNA pathway. Unfortunately, despite multiple attempts, we couldn't obtain viable MCF10A clones that harbor DICER mutation using CRISPR/Cas9-based approaches, and have to settle for siRNA based approach to delete DICER in MCF10A cells. (updated Fig. 3-4, supplementary Fig. 3, 5-6)

3) The ceRNA effect is supposed to happen by competition for miRNA binding, but regular sites are not supposed to reduce the expression level of the miRNA. How do the authors explain the increase in miR-21 levels when using siTGFBI in Fig. 3C, which is abrogated when the TGFBI 3'UTR is used? Especially since mir-21 is not reduced when TGFBI is induced naturally during EMT (Fig. 3B)? Similarly, how is the induction in miR-200c levels in Fig. 4B explained?

Response: This is a very sharp observation. Indeed, the ceRNA effects are not supposed to reduce miRNA expression. However, the system analyzed in our study is very special because the ceRNA interactions are directly coupled with double negative feedback loops regulating EMT. Hence, the observed miRNA expression changes are direct outcomes of the double negative feedback loops between ZEB1-miR-200c and FOXP1-miR-21. Consistent with a ceRNA effect of TGFBI, knocking down TGFBI will release more functional miR-21 molecules without changing the total number of miR-21 molecules, which will in turn lead to a reduced level of FOXP1, which is confirmed by results in Fig 3C. Because FOXP1 and miR-21 form a double negative feedback loop (which we also confirmed in Figure 2), less FOXP1 naturally leads to upregulated expression of miR-21.

As for the reason why miR-21 is not reduced when TGFBI is induced naturally during EMT, it is expected when mechanism other than TGFBI is included into the picture. The expression of miR-21 is regulated simultaneously by multiple mechanisms, including

repression by FOXP1 as shown in Figure 2, induction via TGF-beta stimulation through SMAD based mechanism (Davis et al., Nature. 3;454:56-61), and other undocumented mechanisms. Hence the level of miR-21 is the combined effects of multiple mechanisms, and lack of repression when TGFBI is induced during EMT is expected since studies have reported that TGF-beta treatment can strongly induce miR-21 via SMAD based mechanisms, which is also confirmed by our results. This is also consistent with the downregulation of miR-21 when TGFBI is knock down, because other factors influence miR-21 levels are undisrupted while FOXP1 activity is reduced. The upregulation of miR-21 actually highlights the critical role of TGFBI as an ceRNA during EMT, whose effects are indispensable for abolishing the inhibitory effect of miR-21 to induce effective EMT in A549 cells as shown by our analyses.

Finally, the observed upregulation of miR-200c can be explained similarly to miR-21: knocking down FN1 abolished its ceRNA effects, which lead to more functional miR-200c. More functional miR-200c leads to lower level of ZEB1, which in turn de-repress miR-200c expression owing to the double negative feedback loop between ZEB1 and miR-200c, resulting in a higher level of miR-200c.

The coupling of the ceRNA effect with double negative feedback loops can also explain why perturbing miRNA activities through ceRNA, which are generally considered to be less potent than these of TFs, can lead to substantial expression changes of gene in EMT. Previous study demonstrated that the double negative feedback loop in EMT generate bi-stability, a manifestation of which is the switch like hypersensitive response of induced gene expression. Critically, the timing of upregulation of ceRNA activity is aligned with the timing of the switch, which is confirmed by simulation shown in Figure 5. Hence, a relative small effect induced by ceRNA can be readily amplified by the bi-stability switch, resulting in substantial gene expression changes.

Minor comments:

4) The authors claim that cancer cells “typically express lower levels of miRNAs” (Page 4), but as far as I know that are reports for both decreased and increased overall miRNA abundance in tumors. Is there evidence that the overall miRNA abundance in the cells studied is indeed low compared to hepatocytes or other cell types where the ceRNA hypothesis was studied before? This should be cited or shown.

Response: We agree that our original description is biased and have updated the references to reflect the fact that both decreased and increased overall miRNA abundance in tumors are reported.

More importantly, experimental data confirmed that the overall miRNA abundance in the cells used in our study is indeed lower compared to hepatocytes or other cell types. In the cancerous A549 cells, miRNA-seq data demonstrated that miR-21 is the highest expressing miRNA. Critically, absolute quantification showed that the absolute count of miR-21 molecules in resting A549 cells is 2313 ± 200 copies/cell, and 4220 ± 384 copies/cell at 96h after TGF-beta stimulation, which is about two order of magnitudes lower than the highest expressing miRNA in hepatocytes (miR-122, 1.2×10^5 copies/cell). Moreover, similar to reported miRNA expression landscapes, the overall miRNA expression in A549 cells is dominated by a few highly expressed miRNAs and the vast majority of miRNAs are expressed at very low level. In fact, in A549 cells the total number of reads mapped to miR-21 alone is about 20% of all mapped reads, which put the total number of miRNA molecules in A549 cells at about 10,000 copies/cell (assuming a linear relationship between the number of mapped miRNA-SEQ reads and the absolute count of corresponding molecules). Using a similar strategy, we estimates that there are about 4.4×10^5 miRNA molecules in hepatocytes. Thus experimental data suggested that the total number of miRNAs in A549 cell is about 44 times lower than the total number of miRNAs in hepatocytes. Although the mechanism of broad miRNA downregulation in A549 cells is not clear, MYC, the oncogene that is highly overexpressed in A549 cells (Fukazawa et al. Inhibition of Myc effectively targets KRAS mutation-positive lung cancer expressing high levels of Myc. *Anticancer Res.* 2010 Oct; 30(10):4193-200.), has been demonstrated to broadly lower miRNA expression (Chang et. al 2008. Widespread microRNA repression by Myc contributes to tumorigenesis. *Nat. Genet.* 40, 43–50.), and maybe the culprit for observed broad downregulation of miRNAs in A549 cells.

A different scenario is observed in the benign MCF10A cells. Briefly, miRNA-seq data shows that miR-200c is the 15th highest expressing miRNA in MCF10A (211,178 rpkm) and its absolute quantification is 1396 ± 240 copies/cell (resting) and 1094 ± 39 copies/cell (96h after TGF-beta stimulation). The highest expressing miRNA in MCF10A is miR-378a-3p at 5,886,078 rpkm. Assuming a linear mapping between absolute quantification and miRNA-seq, we estimate there are about 38,910 miR-378a-3p molecules in MCF10A cells,

and about 2×10^5 total miRNA molecules in MCF10A cells. Consequently, the overall miRNA is about 50% lower than the total number of miRNAs in hepatocytes (4.4×10^5).

The fact that miR-200c is only the 15th highest expressing miRNA in MCF10A cells raises a natural question: whether the other 14 high expressing miRNAs influence the ceRNA interactions mediated by miR-200c. Because the high expressing miRNAs (such as miR-378a-3p) outnumber FN1 by an order of magnitude, if any of these miRNAs also bind to ZEB1, then FN1 will not be able to modulate ZEB1 activity via ceRNA-based mechanism. Critically, the targetScan results demonstrates that none of the 14 high expressing miRNAs binds to ZEB1 and only miR-27-3p binds FN1, suggesting that their high expression has no impact on the ceRNA based regulatory network between ZEB1-miR-200c-FN1. This result is consistent with publications that demonstrate a critical role of miR-200 family in regulating EMT through the ZEB1-miR-200 regulatory axis using MCF10A cell lines (Iliopoulos et al., *Sci Signal.* 2(92): ra62.).

Results of this analysis has been added to the manuscript as supplementary table 1 and 2.

5) Which isoform of FN1 is induced? According to TargetScan there are two isoforms and only one of them is targeted by mir-200.

Response: The isoforms containing miR-200c binding sites are the dominant FN1 isoforms expressed in MCF10A cells. This is evident from RT-qPCR results in Figure 3 where primers targeting the miR-200c region successfully amplified FN1 3'UTR. More precisely, we analyzed several MCF10A RNASEQ data and all data sets confirmed that isoforms containing miR-200c-3p binding sites are the dominant FN1 isoforms expressed in MCF10A cells. Results from an example data set (GSE71862), where 94.2% of FN1 transcripts containing miR-200c-3p binding sites, is listed below:

transcript_id	TPM	IsoPct	miR-200c MRE
ENST00000446046	7.4	31.74	Y
ENST00000443816	5.96	25.57	Y

ENST00000356005	4.09	17.56	Y
ENST00000432072	2.37	10.18	Y
ENST00000456923	1.81	7.77	Y
ENST00000474036	1.02	4.37	N
ENST00000492816	0.15	0.62	Y
ENST00000498719	0.14	0.59	N
ENST00000323926	0.13	0.56	Y
ENST00000473614	0.09	0.41	N
ENST00000496542	0.06	0.28	N
ENST00000359671	0.04	0.16	Y
ENST00000426059	0.03	0.15	N
ENST00000336916	0.01	0.05	Y

Reviewer #2 (Remarks to the Author):

In this paper, the authors investigate the importance of ceRNA regulation within the context of EMT. To do so, they characterize two specific examples of ceRNA-miRNA pairs in two different cell lines, and conclude that ceRNA regulation occurs and plays an important role in EMT. Furthermore, evaluation of miRNA and ceRNA stoichiometry provides some support for ceRNA in this specific cancer cell line context.

Comments:

Major:

1. The authors provide some evidence pointing to a relevant ceRNA mediated mechanism modulating the dynamics of EMT. However, like numerous previous studies they fall short of proving this genetically. The authors should capitalize on their cell system and mutate the miR-21 and miR-200c binding sites (by single point mutations) in the TGFBI and FN1 genes, respectively, using readily available genome editing approaches and test if this affects EMT. These are key experiments that that are essential for the conclusions of the manuscript.

Response: We agree with review that genetic experiments are critical to support our results, which is also pointed out by reviewer #1.

As described in our response to comment #2 of reviewer #1, we have mutated the binding sites by CRISPR/Cas9 and performed new experiments, which confirmed that FN1 and TGFBI indeed demonstrate ceRNA effect during EMT.

2. The authors, throughout the manuscript, measure EMT markers and study cell lines in surrogate assays, without determining of whether the postulated ceRNA-miRNA-target gene networks are relevant in in vivo conditions (i.e. metastasizing and non-metastasizing mouse or human tumors).

Response: We agree that in vivo data will substantially support our models. To this aim, we performed survival analyses of three independent lung cancer data sets. Critically, in all analyzed data sets, patients expressing a higher level of TGFBI consistently demonstrate a worse clinical outcome with a higher risk of relapse-free survival, supporting the notion that overexpression TGFBI promotes EMT and metastasis. We didn't perform similar analyses with FN1 because the miR-200s families of miRNAs are routinely downregulated in breast cancer, rendering the FN1-miR-200c axis irrelevant in breast cancer. The survival analyses results have been added into Figure 3H.

3. A recent study by Denzler et al. (Mol Cell. 64: 565–579, 2016) found that cooperative binding of proximal sites for the same or different miRNAs can increase potency and therefore make ceRNA effects more likely. If such closely-spaced sites exist should be investigated and discussed.

Response: This is a very relevant point to examine. Unfortunately, no such proximal sites exist in the analyzed mRNAs. Although pictar predicted two miR-200c binding sites in the 3'UTR of FN1, these two predicted sites are > 200 bp apart. Critically, only one of the two predicted site is functional. The functional site has been previously experimentally validated and its function also supported by our data. The other site is not functional because knocking out the site by CRISPR/Cas9 has no impact on its putative ceRNA effects (supplementary Figure 9B and 10A). Moreover, predicted secondary structure demonstrated that seed region of the predicted nonfunctional site locates in stem regions, further suggesting that the site is not functional (supplementary Figure 9F).

4. In lines 104-116, the authors state that there is a discrepancy between a long miR-200c half-life and rapid ZEB1/CDH1/CDH2 dynamics, concluding that unknown mechanisms must modulate miR-200c expression in order to resolve this issue. This view seems reductionist and ignores factors other than miR-200c that may regulate ZEB1 and its downstream effectors, such as SNAIL1 (Dave et al., 2011, JBC).

Response: The known factors such as SNAIL1 are already incorporated into the model of Zhang et al., through which we derived our hypothesis. Basically, after considering all known factors, current model still couldn't reconcile the discrepancy between miRNA half-life and observed activities, hence we hypothesize that other unknown mechanism must exist. We have modified the manuscript to clearly state that the model incorporating all known regulators failed to explain the experimental observations and other factors are involved.

5. There is some contradiction related to the point at which FOXP1 reaches equilibrium – is it 48h (line 157) or 96h (line 168)? This is relevant to evaluating the dynamics of the system.

Response: FOXP1 reaches equilibrium at 48h and maintains the same expression level up to 96h. We have revised the manuscript to consistently state that FOXP1 reaches equilibrium at 48h.

6. Does miR-21 regulate other EMT genes? The authors state that miR-21 knockdown leads to enhanced EMT (lines 198-199), but there is no experimental evidence that this is occurring through FOXP1 repression. In order to make this claim, it should be experimentally shown that mutation of the FOXP1 miR-21 binding site prevents enhanced migration/invasion upon miR-21 knockdown.

Response: We have performed additional network enrichment analysis on miR-21 targeted genes documented in mirTarbase (a database of experimentally validated miRNA-target interactions). The results showed that TGF-beta signaling pathway is the main relevant pathway targeted by miR-21. Ten genes from TGF-beta signaling pathway are targeted by miR-21 (Smad7, SP1, TGIF1, BMPR2, GDF5, TGFB1, TGFB2, TGFBR2, MYC, ZFYVE16). Because our data showed that FOXP1 is the main TF upregulated during TGF-

beta induced EMT in A549 cells, we expect FOXP1 to serve as hub to integrate upstream signals. Consequently, we expect all the effects operate through FOXP1.

To experimentally validate that the observed effects operated through FOXP1, we followed reviewer's suggestion and utilized CRISPR/Cas9 to knock the miR-21 binding site in FOXP1 3'UTR. We then performed new experiments to confirm that the mutation of the FOXP1 miR-21 binding site indeed prevents enhanced migration/invasion upon miR-21 knockdown. The results have been incorporated into the manuscript as supplementary Figure 3.

7. There is no explanation as to why different cell lines are used to study the miR-21-TGFBI and miR-200c-FN1 regulatory networks. Expression levels of both networks in both cell lines would be informative to help understand stoichiometric requirements for such ceRNA events to occur, or explain why they do not occur under specific circumstances. Furthermore, the authors should discuss the evidence that the observed expression changes/dynamics are relevant for certain primary (as opposed to cell lines) cancer types.

Response: Regulators of EMT demonstrate strong tissue specificity. Consequently, the molecular circuits controlling EMT is very different in A549 and MCF10A cells. Specifically, ZEB1/SNAIL and miR-200c are expressed at very low levels in A549 cells (Chang et. al Synergistic action of master transcription factors controls epithelial-to-mesenchymal transition. *Nucleic Acids Research*, and miR-seq data reported in this study). Similarly, miR-21 is expressed at very low levels in MCF10A cells (supplementary table 1), and FOXP1 is expressed at low level (3.55 TPM, Comaills V et al., *Genomic Instability Is Induced by Persistent Proliferation of Cells Undergoing Epithelial-to-Mesenchymal Transition*, *Cell Rep*, 2016 17(10):2632-2647) and is actually downregulated during EMT in MCF10A cells (Javaid et al. *Dynamic chromatin modification sustains epithelial-mesenchymal transition following inducible expression of Snail-1*. *Cell Rep* 2013 Dec 26;5(6):1679-89., expression fold change of FOXP1: 3h:0.80, 6h:0.85, 12h:0.68, 24h:0.92, 72:0.45, 120:0.28). We have added the expression levels of relevant molecules in the manuscript. Despite the difference in molecules involved, they share common network wiring principles, which involve double negative feedback loops between TF-miRNAs and our newly discovered ceRNAs. Thus using different cellular models help to strength the notion that there are common principles underlying EMT. For the relevance with primary tumors, we have performed survival

analysis of lung cancer to illustrate the potential impact of ceRNA components of EMT regulatory circuits on the survival outcome of cancer patients (Figure 3H).

Minor:

1. Lines 292-301 should be rewritten more clearly, as it is not clear to what “transcriptional repression” refers (presumably to ZEB1 repression of miR-200c?), nor how the time points were selected (how is the point of “maximal ceRNA potency” calculated?).

Response: “transcriptional repression” indeed refers to ZEB1 repression of miR-200c, and we have modify the text to clarify this issue. The time point of “maximal ceRNA potency” is selected using the following criteria. For A549, the time point is selected when TGFBI, the putative ceRNA, reached maximal expression value. For MCF10A, the time point is selected such that FN1 is with highest expression subjected to the constrain that miR-200c is not substantially downregulated. The manuscript has been modified to incorporate these details.

Reviewer #3 (Remarks to the Author):

The manuscript by Du et al reports that TGFbeta and Fibronectin mRNA can act as ceRNAs in the control of EMT in A549 lung adenocarcinoma cells and MCF10A immortalized mammary cells respectively. The authors show that the level of expression of miR-21 and miR-200C are modulated in the presence of their corresponding ceRNA. The kinetics model initially developed by Tian et al., (2015) has been modified to introduce the contribution of the ceRNA. The findings are of interest to further unravel mechanisms driving epithelial cell plasticity.

Remarks

A 549 adenocarcinoma cells are known to be morphologically very heterogeneous with clusters of epithelial-like cells scattered among cells with a pronounced mesenchymal morphology. MCF10A has a basal-like phenotype as most normal-like mammary epithelial cells and again raises issues in terms of epithelial mesenchymal transition. The authors should at least discuss this issue.

Response: These issues are critical for successfully utilizing A549 and MCF10A as cellular models for EMT. We didn't discuss these because we are following the practice of the

community to keep materials & methods concise. We are glad that the reviewer bring this up and have added detailed description and discussion into the manuscript in results and materials & methods sections.

Briefly, A549 cells in normal culture undergo spontaneous EMT largely owing to the TGF-beta presented in FBS. Studies have shown that there are about 1-2ng/ml TGF-beta in 10% FBS (Danielpour et al., Growth Factors 2, 61.; Oida et al., Journal of Immunological Methods 362, 31), which is sufficient to induced partial EMT according to Zhang et al., (Sci Signal 7, 345) and according to our observations. Hence, a serum starvation phase is standard to bring A549 cells to epithelial morphology before the addition of TGF-beta.

MCF10A cells demonstrate additional layer of complexity because studies have shown that MCF10A cells undergo spontaneous EMT when cultured at low cell density (Sarrío et al., Cancer Res 68: 989-997.; Maeda et al., J Cell Sci 118: 873-887.). Hence typical practice using MCF10A as cellular model of EMT is to seed MCF10A cells at > 70% confluency.

These technical details have been addressed in the updated manuscript and should help other researchers using A549 or MCF10A as models for EMT.

The result section describing Figure 6 needs to be considerably improved; Each frame should be described accurately. Clearly the reversibility to an epithelia-like morphology is not obvious.

Response: We have split Figure 6 into two separated figures (A549 data in Figure 6, and MCF10A data in Figure 7). In the new figures, each frame is individually labeled and provides a clear and unbiased reference. We have also revised the manuscript and described the results with greater detail and improved accuracy according to the improved figures, which should make the reversibility to an epithelia-like morphology clear and easy to follow.

Reviewers' Comments:

Reviewer #1:

Remarks to the Author:

The authors have extensively addressed the comments from the previous round of review and now provide more compelling evidence that FN1 and TGFBI can act as ceRNAs for miR-200 and miR-21 in the studied conditions. The manuscript has thus improved substantially. I do not request additional experiments, just additional clarifications:

1. The math underlying Figure 3A is not clear. What is the expression of all the other miR-21 targets in these conditions? It appears that their combined abundance is ~2,500 molecules, so 2.5% of the mRNA in the cell, compared to TGFBI which is 6% of the mRNAs in the cell? How was this computed? Can the authors show the actual expression levels of the miR-21 targets in these conditions (e.g., in a table), which will show how they compare to TGFBI and how ">90% of the new MREs were contributed by TGFBI" conclusion is reached? Same for FN1, the authors report that "FN1, represented over 90% of all increased MREs during EMT in MCF10A cells.", but specific nothing is shown to support this claim.

Minor comments:

1. How is TGFBI siRNA affecting miR-21 levels? This should be explained/discussed.
2. Labels in Supplemental Figure 6F appear to be in the wrong order (as the levels of TGFBI are higher in siTGFBI than in siCtrl).

Reviewer #2:

Remarks to the Author:

The authors have performed several genetic experiments (i.e. introducing mutations in the binding sites using CRISPR/Cas9) which confirmed that FN1 and TGFBI can act as ceRNAs and effect EMT in cancer cell lines. These new data are critical and convincing. The authors also have responded adequately to most of my comments.

However, I do disagree with some claims: The authors are at several places not very precise with regard to their conclusions and they overstate the meaning of their results:

Abstract: "These results help to establish the physiological relevance of ceRNA ..." . The authors are showing the PATHOphysiological relevance of two ceRNAs in A549 and MCF10A cells, which are malignant cells in a hypersensitive cancer background and constitute a model for cancer EMT but are NOT a model of physiological EMT (i.e. embryonic development).

In that respect the study by Denzler et al. is relevant since it studies ceRNAs in a physiological state of primary, non dividing differentiated cells. This is more likely to be relevant (in terms of changes in gene expression that can be observed in stressed conditions) rather than being a particular cell type (lines 121-127). Therefore, the authors should refrain from claiming that they provide evidence for a role of ceRNA regulated gene expression in physiological states (lines 64-65, 146-148, 182, 611-612).

The paper by Title et al. (Nature Communications, vol 9, Article number: 4671 (2018)) recently reported on a study in which the miR-200–Zeb1 axis was genetically dissected in mice, revealing its extreme gene dosage and sensitivity for regulating EMT, tumor differentiation and invasion in cancer, but not in normal physiology. This study is very relevant because it highlights the special role of the Zeb1-miR-200 axis in a sensitized oncogenic background which can be influenced by other factors, as nicely shown in this paper, but is likely the main driver. This of course would question that direct coupling of ceRNA with feedback regulatory loops represents a common and universal mode of action for ceRNAs, but rather be restricted to a few pathways.

Reviewer #3:

Remarks to the Author:

The manuscript has been significantly improved. The authors have taken into considerations my criticisms

Reviewer #1 (Remarks to the Author):

The authors have extensively addressed the comments from the previous round of review and now provide more compelling evidence that FN1 and TGFBI can act as ceRNAs for miR-200 and miR-21 in the studied conditions. The manuscript has thus improved substantially. I do not request additional experiments, just additional clarifications:

1. The math underlying Figure 3A is not clear. What is the expression of all the other miR-21 targets in these conditions? It appears that their combined abundance is ~2,500 molecules, so 2.5% of the mRNA in the cell, compared to TGFBI which is 6% of the mRNAs in the cell? How was this computed? Can the authors show the actual expression levels of the miR-21 targets in these conditions (e.g., in a table), which will show how they compare to TGFBI and how “>90% of the new MREs were contributed by TGFBI” conclusion is reached? Same for FN1, the authors report that “FN1, represented over 90% of all increased MREs during EMT in MCF10A cells.”, but specific nothing is shown to support this claim.

Response: Our apologies for missing these information, which are important for understanding the results as pointed out by the reviewer. We have added a new section “Estimating number of MREs using gene expression data” in method to describe the procedures we used to derived number of MREs from gene expression data.

We have also added two new tables showing the expression levels of miR-21 (supplementary table 1) and miR-200c targets (supplementary table 3). Finally, the number of MREs derived from RNASEQ are provided in two new tables (supplementary table 2 and 4 respectively) supporting “FN1, represented over 90% of all increased MREs during EMT in MCF10A cells.” And “>90% of the new MREs were contributed by TGFBI”.

Minor comments:

1. How is TGFBI siRNA affecting miR-21 levels? This should be explained/discussed.

Response: we have edited the manuscript and specifically explained that TGFBI can directly modulate miR-21 activity through the FOXP1-miR-21 double negative feedback loop.

2. Labels in Supplemental Figure 6F appear to be in the wrong order (as the levels of TGFBI are higher in siTGFBI than in siCtrl).

Response: the labels have been corrected.

Reviewer #2 (Remarks to the Author):

The authors have performed several genetic experiments (i.e. introducing mutations in the binding sites using CRISPR/Cas9) which confirmed that FN1 and TGFBI can act as ceRNAs and effect EMT in cancer cell lines. These new data are critical and convincing. The authors also have responded adequately to most of my comments.

However, I do disagree with some claims: The authors are at several places not very precise with regard to their conclusions and they overstate the meaning of their results:

Abstract: "These results help to establish the physiological relevance of ceRNA ..." . The authors are showing the PATHOphysiological relevance of two ceRNAs in A549 and MCF10A cells, which are malignant cells in a hypersensitive cancer background and constitute a model for cancer EMT but are NOT a model of physiological EMT (i.e. embryonic development).

In that respect the study by Denzler et al. is relevant since it studies ceRNAs in a physiological state of primary, non dividing differentiated cells. This is more likely to be relevant (in terms of changes in gene expression that can be observed in stressed conditions) rather than being a particular cell type (lines 121-127). Therefore, the authors should refrain from claiming that they provide evidence for a role of ceRNA regulated gene expression in physiological states (lines 64-65, 146-148, 182, 611-612).

Response: We agree that our current results should be confined in the cancer EMT category and these more speculative statements have been revised according to reviewer's suggestions. Specifically, we have edited all the descriptions pointed out by the reviewer and revised them such that now all statements are restricted to EMT in cancer.

The paper by Title et al. (Nature Communications, vol 9, Article number: 4671 (2018)) recently reported on a study in which the miR-200–Zeb1 axis was genetically dissected in mice, revealing its extreme gene dosage and sensitivity for regulating EMT, tumor differentiation and invasion in cancer, but not in normal physiology. This study is very relevant because it highlights the special role of the Zeb1-miR-200 axis in a sensitized oncogenic background which can be influenced by other factors, as nicely shown in this

paper, but is likely the main driver. This of course would question that direct coupling of ceRNA with feedback regulatory loops represents a common and universal mode of action for ceRNAs, but rather be restricted to a few pathways.

Response: We agree that the impact of ceRNA alone is generally limited, and its role is likely to be supportive comparing to TFs. We have edited the manuscript to reflect this change. For example, in discussion we change “ceRNA may represent a universal mode of action for ceRNAs.” to “suggest that ceRNAs may represent candidate regulators in pathways utilizing TF-miRNA feedback loops.”. We also cited the paper by Title et al. and briefly discussed its relevance to our work.